# Tractable Density Estimation on Learned Manifolds with Conformal Embedding Flows

**Brendan Leigh Ross**
Layer 6 AI
brendan@layer6.ai

**Jesse C. Cresswell**
Layer 6 AI
jesse@layer6.ai

## Abstract

Normalizing flows are generative models that provide tractable density estimation via an invertible transformation from a simple base distribution to a complex target distribution. However, this technique cannot directly model data supported on an unknown low-dimensional manifold, a common occurrence in real-world domains such as image data. Recent attempts to remedy this limitation have introduced geometric complications that defeat a central benefit of normalizing flows: exact density estimation. We recover this benefit with Conformal Embedding Flows, a framework for designing flows that learn manifolds with tractable densities. We argue that composing a standard flow with a trainable conformal embedding is the most natural way to model manifold-supported data. To this end, we present a series of conformal building blocks and apply them in experiments with synthetic and real-world data to demonstrate that flows can model manifold-supported distributions without sacrificing tractable likelihoods.

## 1 Introduction

Deep generative modelling is the task of modelling a complex, high-dimensional data distribution from a sample set. Research has encompassed major approaches such as normalizing flows (NFs) [16, 59], generative adversarial networks (GANs) [23], variational autoencoders (VAEs) [36], autoregressive models [52], energy-based models [18], score-based models [64], and diffusion models [29, 63]. NFs in particular describe a distribution by modelling a change-of-variables mapping to a known base density. This approach provides the unique combination of efficient inference, efficient sampling, and exact density estimation, but in practice generated images have not been as detailed or realistic as those of those of other methods [7, 10, 29, 32, 68].

One limitation of traditional NFs is the use of a base density with the same dimensionality as the data. This stands in contrast to models such as GANs and VAEs, which generate data by sampling from a low-dimensional latent prior and mapping the sample to data space. In many application domains, it is known or commonly assumed that the data of interest lives on a lower-dimensional manifold embedded in the higher-dimensional data space [21]. For example, when modelling images, data samples belong to $[0, 1]^n$, where $n$ is the number of pixels in each image and each pixel has a brightness in the domain $[0, 1]$. However, most points in this data space correspond to meaningless noise, whereas meaningful images of objects lie on a submanifold of dimension $m \ll n$. A traditional NF cannot take advantage of the lower-dimensional nature of realistic images.

There is growing research interest in *injective flows*, which account for unknown manifold structure by incorporating a base density of lower dimensionality than the data space [6, 12, 13, 39, 41]. Flows with low-dimensional latent spaces could benefit from making better use of fewer parameters, being more memory efficient, and could reveal information about the intrinsic structure of the data. Properties of the data manifold, such as its dimensionality or the semantic meaning of latent directions,

can be of interest as well [36, 58]. However, leading injective flow models still suffer from drawbacks including intractable density estimation [6] or reliance on stochastic inverses [13].

In this paper we propose Conformal Embedding Flows (CEFs), a class of flows that use *conformal embeddings* to transform from low to high dimensions while maintaining invertibility and an efficiently computable density. We show how conformal embeddings can be used to learn a lower dimensional data manifold, and we combine them with powerful NF architectures for learning densities. The overall CEF paradigm permits efficient density estimation, sampling, and inference. We propose several types of conformal embedding that can be implemented as composable layers of a flow, including three new invertible layers: the orthogonal $k \times k$ convolution, the conditional orthogonal transformation, and the special conformal transformation. Lastly, we demonstrate their efficacy on synthetic and real-world data.

## 2 Background

### 2.1 Normalizing Flows

In the traditional setting of a normalizing flow [16, 59], an independent and identically distributed sample $\{\mathbf{x}_i\} \subset \mathcal{X} = \mathbb{R}^n$ from an unknown ground-truth distribution with density $p_{\mathbf{x}}^*(\mathbf{x})$ is used to learn an approximate density $p_{\mathbf{x}}(\mathbf{x})$ via maximum likelihood estimation. The approximate density is modelled using a diffeomorphism $\mathbf{f} : \mathcal{Z} \to \mathcal{X}$ which maps a base density $p_{\mathbf{z}}(\mathbf{z})$ over the space $\mathcal{Z} = \mathbb{R}^n$, typically taken to be a multivariate normal, to $p_{\mathbf{x}}(\mathbf{x})$ via the change of variables formula

$$p_{\mathbf{x}}(\mathbf{x}) = p_{\mathbf{z}}\left(\mathbf{f}^{-1}(\mathbf{x})\right) \left|\det \mathbf{J}_{\mathbf{f}}\left(\mathbf{f}^{-1}(\mathbf{x})\right)\right|^{-1}, \tag{1}$$

where $\mathbf{J}_{\mathbf{f}}(\mathbf{z})$ is the Jacobian matrix of $\mathbf{f}$ at the point $\mathbf{z}$. In geometric terms, the probability mass in an infinitesimal volume $\mathbf{dz}$ of $\mathcal{Z}$ must be preserved in the volume of $\mathcal{X}$ corresponding to the image $\mathbf{f}(\mathbf{dz})$, and the magnitude of the Jacobian determinant is exactly what accounts for changes in the coordinate volume induced by $\mathbf{f}$. By parameterizing classes of diffeomorphisms $\mathbf{f}_\theta$, the flow model can be fitted via maximum likelihood on the training data $\{\mathbf{x}_i\} \subset \mathcal{X}$. Altogether, the three following operations must be tractable: sampling with $\mathbf{f}(\mathbf{z})$, inference with $\mathbf{f}^{-1}(\mathbf{x})$, and density estimation with the $|\det \mathbf{J}_{\mathbf{f}}(\mathbf{z})|$ factor. To scale the model, one can compose many such layers $\mathbf{f} = \mathbf{f}_k \circ \cdots \circ \mathbf{f}_1$, and the $|\det \mathbf{J}_{\mathbf{f}_i}(\mathbf{z})|$ factors multiply in Eq. (1).

Generally, there is no unifying way to parameterize an arbitrary bijection satisfying these constraints. Instead, normalizing flow research has progressed by designing new and more expressive component bijections which can be parameterized, learned, and composed. In particular, progress has been made by designing invertible layers whose Jacobian determinants are tractable by construction. A significant theme has been to structure flows to have a triangular Jacobian [16, 59, 37, 17]. Kingma and Dhariwal [35] introduced invertible $1 \times 1$ convolution layers for image modelling; these produce block-diagonal Jacobians whose blocks are parameterized in a $PLU$-decomposition, so that the determinant can be computed in $\mathcal{O}(c)$, the number of input channels. See Papamakarios et al. [54] for a thorough survey of normalizing flows.

### 2.2 Injective Flows

The requirement that $\mathbf{f}$ be a diffeomorphism fixes the dimensionality of the latent space. In turn, $p_{\mathbf{x}}(\mathbf{x})$ must have full support over $\mathcal{X}$, which is problematic when the data lies on a submanifold $\mathcal{M} \subset \mathcal{X}$ with dimension $m < n$. Dai and Wipf [14] observed that if a probability model with full support is fitted via maximum likelihood to such data, the estimated density can converge towards infinity on $\mathcal{M}$ while ignoring the true data density $p^*(\mathbf{x})$ entirely. Behrmann et al. [5] point out that analytically invertible neural networks can become numerically non-invertible, especially when the effective dimensionality of data and latents are mismatched. Correctly learning the data manifold along with its density may circumvent these pathologies.

Injective flows seek to learn an explicitly low-dimensional support by reducing the dimensionality of the latent space and modelling the flow as a *smooth embedding*, or an injective function which is

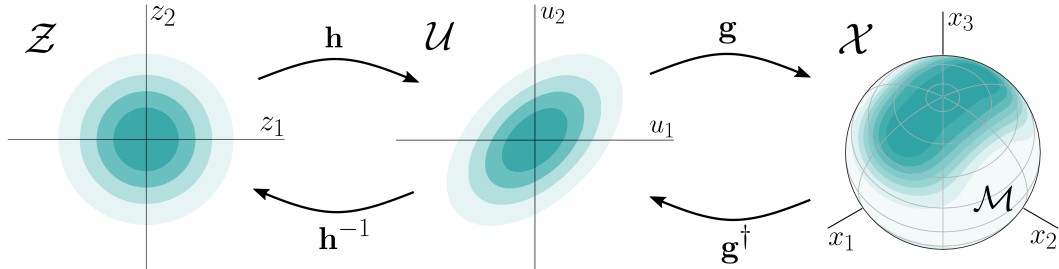

Figure 1: A normalized base density in the $\mathcal{Z}$ space is mapped by a bijective flow $\mathbf{h}$ to a more complicated density in $\mathcal{U}$. The injective component $\mathbf{g}$ maps this density onto a manifold $\mathcal{M}$ in $\mathcal{X}$. For inference, data points from $\mathcal{M}$ follow the reverse path through $\mathbf{g}^\dagger$ and $\mathbf{h}^{-1}$ to the latent space $\mathcal{Z}$ where their densities can be evaluated and combined with the determinant factors in Eq. (4).

diffeomorphic to its image[1]. This case can be accommodated with a generalized change of variables formula for densities as follows [22].

Let $\mathbf{g} : \mathcal{U} \to \mathcal{X}$ be a smooth embedding from a latent space $\mathcal{U}$ onto the data manifold $\mathcal{M} \subset \mathcal{X}$. That is, $\mathcal{M} = \mathbf{g}(\mathcal{U})$ is the range of $\mathbf{g}$. Accordingly, $\mathbf{g}$ has a left-inverse[2] $\mathbf{g}^\dagger : \mathcal{X} \to \mathcal{U}$ which is smooth on $\mathcal{M}$ and satisfies $\mathbf{g}^\dagger(\mathbf{g}(\mathbf{u})) = \mathbf{u}$ for all $\mathbf{u} \in \mathcal{U}$. Suppose $p_\mathbf{u}(\mathbf{u})$ is a density on $\mathcal{U}$ described using coordinates $\mathbf{u}$. The same density can be described in the ambient space $\mathcal{X}$ using $\mathbf{x}$ coordinates by pushing it through $\mathbf{g}$.

The quantity that accounts for changes in the coordinate volume at each point $\mathbf{u}$ is $\sqrt{\det\left[\mathbf{J}_\mathbf{g}^T(\mathbf{u})\mathbf{J}_\mathbf{g}(\mathbf{u})\right]}$, where the Jacobian $\mathbf{J}_\mathbf{g}$ is now a $n \times m$ matrix [43]. Hence, using the shorthand $\mathbf{u} = \mathbf{g}^\dagger(\mathbf{x})$, the generalized change of variables formula defined for $\mathbf{x} \in \mathcal{M}$ can be written

$$p_\mathbf{x}(\mathbf{x}) = p_\mathbf{u}(\mathbf{u}) \left|\det\left[\mathbf{J}_\mathbf{g}^T(\mathbf{u})\mathbf{J}_\mathbf{g}(\mathbf{u})\right]\right|^{-\frac{1}{2}}. \tag{2}$$

While $\mathbf{g}$ describes how the data manifold $\mathcal{M}$ is embedded in the larger ambient space, the mapping $\mathbf{g}$ alone may be insufficient to represent a normalized base density. As before, it is helpful to introduce a latent space $\mathcal{Z}$ of dimension $m$ along with a diffeomorphism $\mathbf{h} : \mathcal{Z} \to \mathcal{U}$ representing a bijective NF between $\mathcal{Z}$ and $\mathcal{U}$ [6]. Taking the overall injective transformation $\mathbf{g} \circ \mathbf{h}$ and applying the chain rule $\mathbf{J}_{\mathbf{g}\circ\mathbf{h}} = \mathbf{J}_\mathbf{g}\mathbf{J}_\mathbf{h}$ simplifies the determinant in Eq. (2) since the outer Jacobian $\mathbf{J}_\mathbf{h}$ is square,

$$\det\left[\mathbf{J}_\mathbf{h}^T\mathbf{J}_\mathbf{g}^T\mathbf{J}_\mathbf{g}\mathbf{J}_\mathbf{h}\right] = (\det\mathbf{J}_\mathbf{h})^2 \det\left[\mathbf{J}_\mathbf{g}^T\mathbf{J}_\mathbf{g}\right]. \tag{3}$$

Finally, writing $\mathbf{z} = \mathbf{h}^{-1}(\mathbf{u})$, the data density is modelled by

$$p_\mathbf{x}(\mathbf{x}) = p_\mathbf{z}(\mathbf{z}) \left|\det\mathbf{J}_\mathbf{h}(\mathbf{z})\right|^{-1} \left|\det\left[\mathbf{J}_\mathbf{g}^T(\mathbf{u})\mathbf{J}_\mathbf{g}(\mathbf{u})\right]\right|^{-\frac{1}{2}}, \tag{4}$$

with the entire process depicted in Fig. 1.

Generating samples from $p_\mathbf{x}(\mathbf{x})$ is simple; a sample $\mathbf{z} \sim p_\mathbf{z}(\mathbf{z})$ is drawn from the base density and passed through $\mathbf{g} \circ \mathbf{h}$. Inference on a data sample $\mathbf{x} \sim p_\mathbf{x}(\mathbf{x})$ is achieved by passing it through $\mathbf{h}^{-1} \circ \mathbf{g}^\dagger$, evaluating the density according to $p_\mathbf{z}(\mathbf{z})$, and computing both determinant factors.

Notably, the learned density $p_\mathbf{x}(\mathbf{x})$ only has support on a low-dimensional subset $\mathcal{M}$ of $\mathcal{X}$, as per the manifold hypothesis. This formulation leads the learned manifold $\mathcal{M}$ to be diffeomorphic to Euclidean space, which can cause numerical instability when the data's support differs in topology [11], but we leave this issue to future work.

In practice, there will be off-manifold points during training or if $g(\mathbf{u})$ cannot perfectly fit the data, in which case the model's log-likelihood will be $-\infty$. Cunningham et al. [13] remedy this by adding

---

[1]Throughout this work we use "embedding" in the topological sense: a function which describes how a low-dimensional space can sit inside a high-dimensional space. This is not to be confused with other uses for the term in machine learning, namely a low-dimensional representation of high-dimensional or discrete data.

[2]$\dagger$ denotes a left-inverse function, not necessarily the matrix pseudoinverse.

an off-manifold noise term to the model, but inference requires a stochastic inverse, and the model must be optimized using an ELBO-like objective. Other work [6, 9, 39] has projected data to the manifold via $\mathbf{g} \circ \mathbf{g}^{\dagger}$ prior to computing log-likelihoods and optimized $\mathbf{g}$ using the reconstruction loss $\mathbb{E}_{\mathbf{x} \sim p_{\mathbf{x}}^*} \|\mathbf{x} - \mathbf{g}(\mathbf{g}^{\dagger}(\mathbf{x}))\|^2$. We prove in App. A that minimizing the reconstruction loss brings the learned manifold into alignment with the data manifold.

When computing log-likelihoods, the determinant term $\log \det \left[\mathbf{J}_{\mathbf{g}}^T \mathbf{J}_{\mathbf{g}}\right]$ presents a computational challenge. Kumar et al. [41] maximize it using an approximate lower bound, while Brehmer and Cranmer [6] and Kothari et al. [39] circumvent its computation altogether by only maximizing the other terms in the log-likelihood. In concurrent work, Caterini et al. [9] optimize injective flows using a stochastic estimate of the log-determinant's gradient. They are also able to optimize $\log \det \left[\mathbf{J}_{\mathbf{g}}^T \mathbf{J}_{\mathbf{g}}\right]$ exactly for smaller datasets, but this procedure involves the explicit construction of $\mathbf{J}_{\mathbf{g}}$, which would be memory-intensive to scale to larger data such as CelebA. In line with research to build expressive bijective flows where $\det \mathbf{J}_{\mathbf{f}}$ is tractable, our work focuses on designing and parameterizing injective flows where $\log \det \left[\mathbf{J}_{\mathbf{g}}^T \mathbf{J}_{\mathbf{g}}\right]$ as a whole is efficiently computable. In contrast to past injective flow models, our approach allows for straightforward evaluation and optimization of $\log \det \left[\mathbf{J}_{\mathbf{g}}^T \mathbf{J}_{\mathbf{g}}\right]$ in the same way standard NFs do for $\log |\det \mathbf{J}_{\mathbf{f}}|$. As far as we can find, ours is the first approach to make this task tractable at scale.

## 3  Conformal Embedding Flows

In this section we propose Conformal Embedding Flows (CEFs) as a method for learning the low-dimensional manifold $\mathcal{M} \subset \mathcal{X}$ and the probability density of the data on the manifold.

Modern bijective flow work has produced tractable $\log |\det \mathbf{J}_{\mathbf{f}}|$ terms by designing layers with triangular Jacobians [16, 17]. For injective flows, the combination $\mathbf{J}_{\mathbf{g}}^T \mathbf{J}_{\mathbf{g}}$ is symmetric, so it is triangular if and only if it is diagonal. In turn, $\mathbf{J}_{\mathbf{g}}^T \mathbf{J}_{\mathbf{g}}$ being diagonal is equivalent to $\mathbf{J}_{\mathbf{g}}$ having orthogonal columns. While this restriction is feasible for a single layer $\mathbf{g}$, it is not composable. If $\mathbf{g}_1$ and $\mathbf{g}_2$ are both smooth embeddings whose Jacobians have orthogonal columns, it need not follow that $\mathbf{J}_{\mathbf{g}_2 \circ \mathbf{g}_1}$ has orthogonal columns. Additionally, since the Jacobians are not square the determinant in Eq. (2), $\det \left[\mathbf{J}_{\mathbf{g}_1}^T \mathbf{J}_{\mathbf{g}_2}^T \mathbf{J}_{\mathbf{g}_2} \mathbf{J}_{\mathbf{g}_1}\right]$, cannot be factored into a product of individually computable terms as in Eq. (3). To ensure composability we propose enforcing the slightly stricter criterion that each $\mathbf{J}_{\mathbf{g}}^T \mathbf{J}_{\mathbf{g}}$ be a scalar multiple of the identity. This is precisely the condition that $\mathbf{g}$ is a conformal embedding.

Formally, $\mathbf{g} : \mathcal{U} \to \mathcal{X}$ is a *conformal embedding* if it is a smooth embedding whose Jacobian satisfies

$$\mathbf{J}_{\mathbf{g}}^T(\mathbf{u})\mathbf{J}_{\mathbf{g}}(\mathbf{u}) = \lambda^2(\mathbf{u})\mathbf{I}_m \, , \tag{5}$$

where $\lambda : \mathcal{U} \to \mathbb{R}$ is a smooth non-zero scalar function, the *conformal factor* [43]. In other words, $\mathbf{J}_{\mathbf{g}}$ has orthonormal columns up to a smoothly varying non-zero multiplicative constant. Hence $\mathbf{g}$ locally preserves angles.

From Eq. (5) it is clear that conformal embeddings naturally satisfy our requirements as an injective flow. In particular, let $\mathbf{g} : \mathcal{U} \to \mathcal{X}$ be a conformal embedding and $\mathbf{h} : \mathcal{Z} \to \mathcal{U}$ be a standard normalizing flow model. The injective flow model $\mathbf{g} \circ \mathbf{h} : \mathcal{Z} \to \mathcal{X}$ satisfies

$$p_{\mathbf{x}}(\mathbf{x}) = p_{\mathbf{z}}(\mathbf{z}) \left|\det \mathbf{J}_{\mathbf{h}}(\mathbf{z})\right|^{-1} \lambda^{-m}(\mathbf{u}) \, . \tag{6}$$

We call $\mathbf{g} \circ \mathbf{h}$ a Conformal Embedding Flow.

CEFs provide a new way to coordinate the training dynamics of the model's manifold and density. It is important to note that not all parameterizations $\mathbf{g}$ of the learned manifold are equally suited to density estimation [9]. Prior injective flow models [6, 39] have been trained *sequentially* by first optimizing $\mathbf{g}$ using the reconstruction loss $\mathbb{E}_{\mathbf{x} \sim p_{\mathbf{x}}^*} \|\mathbf{x} - \mathbf{g}(\mathbf{g}^{\dagger}(\mathbf{x}))\|^2$, then training $\mathbf{h}$ for maximum likelihood with $\mathbf{g}$ fixed. This runs the risk of initializing the density $p_{\mathbf{u}}(\mathbf{u})$ in a configuration that is challenging for $\mathbf{h}$ to learn. Brehmer and Cranmer [6] also alternate training $\mathbf{g}$ and $\mathbf{h}$, but this does not prevent $\mathbf{g}$ from converging to a poor configuration for density estimation. Unlike previous injective flows, CEFs have tractable densities, which allows $\mathbf{g}$ and $\mathbf{h}$ to be trained *jointly* by optimizing the loss function

$$\mathcal{L} = \mathbb{E}_{\mathbf{x} \sim p_{\mathbf{x}}^*} \left[-\log p_{\mathbf{x}}(\mathbf{x}) + \alpha \|\mathbf{x} - \mathbf{g}(\mathbf{g}^{\dagger}(\mathbf{x}))\|^2\right] \, . \tag{7}$$

This mixed loss provides more flexibility in how $\mathbf{g}$ is learned, and is unique to our model because it is the first for which $\log p_{\mathbf{x}}(\mathbf{x})$ is tractable.

## 3.1 Designing Conformal Embedding Flows

Having established the model's high-level structure and training objective, it remains for us to design conformal embeddings $\mathbf{g}$ which are capable of representing complex data manifolds. For $\mathbf{g}$ to be useful in a CEF we must be able to sample with $\mathbf{g}(\mathbf{u})$, perform inference with $\mathbf{g}^\dagger(\mathbf{x})$, and compute the conformal factor $\lambda(\mathbf{u})$. In general, there is no unifying way to parameterize the entire family of such conformal embeddings (see App. B for more discussion). As when designing standard bijective flows, we can only identify subfamilies of conformal embeddings which we parameterize and compose to construct an expressive flow. To this end, we work with conformal building blocks $\mathbf{g}_i : \mathcal{U}_{i-1} \to \mathcal{U}_i$ (where $\mathcal{U}_0 = \mathcal{U}$ and $\mathcal{U}_k = \mathcal{X}$), which we compose to produce the full conformal embedding $\mathbf{g}$:

$$\mathbf{g} = \mathbf{g}_k \circ \cdots \circ \mathbf{g}_1 . \tag{8}$$

In turn, $\mathbf{g}$ is conformal because

$$\mathbf{J}_\mathbf{g}^T \mathbf{J}_\mathbf{g} = \left( \mathbf{J}_{\mathbf{g}_1}^T \cdots \mathbf{J}_{\mathbf{g}_k}^T \right) \left( \mathbf{J}_{\mathbf{g}_k} \cdots \mathbf{J}_\mathbf{g} \right) = \lambda_1^2 \cdots \lambda_k^2 \mathbf{I}_m . \tag{9}$$

Our goal in the remainder of this section is to design classes of conformal building blocks which can be parameterized and learned in a CEF.

### 3.1.1 Conformal Embeddings from Conformal Mappings

Consider the special case where the conformal embedding maps between Euclidean spaces $\mathcal{U} \subseteq \mathbb{R}^d$ and $\mathcal{V} \subseteq \mathbb{R}^d$ of the same dimension[3]. In this special case $\mathbf{g}_i$ is called a *conformal mapping*. Liouville's theorem [27] states that any conformal mapping can be expressed as a composition of translations, orthogonal transformations, scalings, and inversions, which are defined in Table 1 (see App. B.1 for details on conformal mappings). We created conformal embeddings primarily by composing these layers. Zero-padding [6] is another conformal embedding, with Jacobian $\mathbf{J}_\mathbf{g} = \left( \mathbf{I}_m \ 0 \right)^T$, and can be interspersed with conformal mappings to provide changes in dimensionality.

Table 1: Conformal Mappings

| TYPE | FUNCTIONAL FORM | PARAMS | INVERSE | $\lambda(\mathbf{u})$ |
|---|---|---|---|---|
| Translation | $\mathbf{u} \mapsto \mathbf{u} + \mathbf{a}$ | $\mathbf{a} \in \mathbb{R}^d$ | $\mathbf{v} \mapsto \mathbf{v} - \mathbf{a}$ | $1$ |
| Orthogonal | $\mathbf{u} \mapsto \mathbf{Q}\mathbf{u}$ | $\mathbf{Q} \in O(d)$ | $\mathbf{v} \mapsto \mathbf{Q}^T \mathbf{v}$ | $1$ |
| Scaling | $\mathbf{u} \mapsto \lambda \mathbf{u}$ | $\lambda \in \mathbb{R}$ | $\mathbf{v} \mapsto \lambda^{-1} \mathbf{v}$ | $\lambda$ |
| Inversion | $\mathbf{u} \mapsto \mathbf{u}/\|\mathbf{u}\|^2$ | | $\mathbf{v} \mapsto \mathbf{v}/\|\mathbf{v}\|^2$ | $\|\mathbf{u}\|^{-2}$ |
| SCT | $\mathbf{u} \mapsto \frac{\mathbf{u} - \|\mathbf{u}\|^2 \mathbf{b}}{1 - 2\mathbf{b}\cdot\mathbf{u} + \|\mathbf{b}\|^2 \|\mathbf{u}\|^2}$ | $\mathbf{b} \in \mathbb{R}^d$ | $\mathbf{v} \mapsto \frac{\mathbf{v} + \|\mathbf{v}\|^2 \mathbf{b}}{1 + 2\mathbf{b}\cdot\mathbf{v} + \|\mathbf{b}\|^2 \|\mathbf{v}\|^2}$ | $1 - 2\mathbf{b}\cdot\mathbf{u} + \|\mathbf{b}\|^2 \|\mathbf{u}\|^2$ |

Stacking translation, orthogonal transformation, scaling, and inversion layers is sufficient to learn any conformal mapping in principle. However, the inversion operation is numerically unstable, so we replaced it with the *special conformal transformation* (SCT), a transformation of interest in conformal field theory [15]. It can be understood as an inversion, followed by a translation by $-\mathbf{b}$, followed by another inversion. In contrast to inversions, SCTs have a continuous parameter and include the identity when this parameter is set to 0.

The main challenge to implementing conformal mappings was writing trainable orthogonal layers. We parameterized orthogonal transformations in two different ways: by using Householder matrices [67], which are cheaply parameterizable and easy to train, and by using GeoTorch, the API provided by [44], which parameterizes the special orthogonal group by taking the matrix exponential of skew-symmetric matrices. GeoTorch also provides trainable non-square matrices with orthonormal columns, which are conformal embeddings (not conformal mappings) and which we incorporate to change the data's dimensionality.

To scale orthogonal transformations to image data, we propose a new invertible layer: the *orthogonal $k \times k$ convolution*. In the spirit of the invertible $1 \times 1$ convolutions of Kingma and Dhariwal [35], we

---

[3]We consider conformal mappings between spaces of dimension $d > 2$. Conformal mappings in $d = 2$ are much less constrained, while the case $d = 1$ is trivial since there is no notion of an angle.

note that a $k \times k$ convolution with stride $k$ has a block diagonal Jacobian. The Jacobian is orthogonal if and only if these blocks are orthogonal. It suffices then to convolve the input with a set of filters that together form an orthogonal matrix. Moreover, by modifying these matrices to be non-square with orthonormal columns (in practice, reducing the filter count), we can provide conformal changes in dimension. It is also worth noting that these layers can be inverted efficiently by applying a transposed convolution with the same filter, while a standard invertible $1 \times 1$ convolution requires a matrix inversion. This facilitates quick forward and backward passes when optimizing the model's reconstruction loss.

### 3.1.2 Piecewise Conformal Embeddings

To make the embeddings more expressive, the conformality condition on $\mathbf{g}$ can be relaxed to the point of being conformal *almost everywhere*. Formally, the latent spaces $\mathcal{Z}$ and $\mathcal{U}$ are redefined as $\mathcal{Z} = \{\mathbf{z} : \mathbf{g} \text{ is conformal at } \mathbf{h}(\mathbf{z})\}$ and $\mathcal{U} = \mathbf{h}(\mathcal{Z})$. Then $\mathbf{g}$ remains a conformal embedding on $\mathcal{U}$, and as long as $\{\mathbf{x} : \mathbf{g} \text{ is nonconformal at } \mathbf{g}^\dagger(\mathbf{x})\}$ also has measure zero, this approach poses no practical problems. Note that the same relaxation is performed implicitly with the diffeomorphism property of standard flows when rectifier nonlinearities are used in coupling layers [17] and can be justified by generalizing the change of variables formula [38].

Table 2: Piecewise Conformal Embeddings

| TYPE | FUNCTIONAL FORM | PARAMS | LEFT INVERSE | $\lambda(\mathbf{u})$ |
|---|---|---|---|---|
| Conformal ReLu | $\mathbf{u} \mapsto \mathrm{ReLU} \begin{bmatrix} \mathbf{Qu} \\ -\mathbf{Qu} \end{bmatrix}$ | $\mathbf{Q} \in O(d)$ | $\begin{bmatrix} \mathbf{v}_1 \\ \mathbf{v}_2 \end{bmatrix} \mapsto \mathbf{Q}^T (\mathbf{v}_1 - \mathbf{v}_2)$ | 1 |
| Conditional Orthogonal | $\mathbf{u} \mapsto \begin{cases} \mathbf{Q}_1\mathbf{u} & \text{if } \|\mathbf{u}\| < 1 \\ \mathbf{Q}_2\mathbf{u} & \text{if } \|\mathbf{u}\| \geq 1 \end{cases}$ | $\mathbf{Q}_1, \mathbf{Q}_2 \in O(d)$ | $\mathbf{v} \mapsto \begin{cases} \mathbf{Q}_1^T\mathbf{u} & \text{if } \|\mathbf{v}\| < 1 \\ \mathbf{Q}_2^T\mathbf{u} & \text{if } \|\mathbf{v}\| \geq 1 \end{cases}$ | 1 |

We considered the two piecewise conformal embeddings defined in Table 2. Due to the success of ReLU in standard deep neural networks [50], we try a ReLU-like layer that is piecewise conformal. *conformal ReLU* is based on the injective ReLU proposed by Kothari et al. [39]. We believe it to be of general interest as a dimension-changing conformal nonlinearity, but it provided no performance improvements in experiments.

More useful was the *conditional orthogonal* transformation, which takes advantage of the norm-preservation of orthogonal transformations to create an invertible layer. Despite being discontinuous, it provided a substantial boost in reconstruction ability on image data. The idea behind this transformation can be extended to the other parameterized mappings in Table 1. For each type of mapping we can identify hypersurfaces in $\mathbb{R}^n$ such that each hypersurface is mapped back to itself; i.e., each hypersurface is an orbit of its points under the mapping. Applying the same type of conformal mapping piecewise on either side remains an invertible operation as long as trajectories do not cross the hypersurface, and the result is conformal *almost everywhere*. The conditional orthogonal layer was the only example of these that provided performance improvements.

## 4 Related Work

**Flows on prescribed manifolds.** Flows can be developed for Riemannian manifolds $\mathcal{M} \subseteq \mathcal{X}$ which are known in advance and can be defined as the image of some fixed $\phi : \mathcal{U} \to \mathcal{X}$, where $\mathcal{U} \subseteq \mathbb{R}^m$ [22, 48, 54]. In particular, Rezende et al. [60] model densities on spheres and tori with convex combinations of Möbius transformations, which are cognate to conformal mappings. For known manifolds $\phi$ is fixed, and the density's Jacobian determinant factor may be computable in closed form. Our work replaces $\phi$ with a trainable network $g$, but the log-determinant still has a simple closed form.

**Flows on learnable manifolds.** Extending flows to learnable manifolds brings about two main challenges: handling off-manifold points, and training the density on the manifold.

When the distribution is manifold-supported, it will assign zero density to off-manifold points. This has been addressed by adding an off-manifold noise term [12, 13] or by projecting the data onto the manifold and training it with a reconstruction loss [6, 39, 41]. We opt for the latter approach.

Training the density on the manifold is challenging because the log-determinant term is typically intractable. Kumar et al. [41] use a series of lower bounds to train the log-determinant, while Brehmer and Cranmer [6] and Kothari et al. [39] separate the flow into two components and train only the low-dimensional component. Caterini et al. [9] maximize log-likelihood directly by either constructing the embedding's Jacobian explicitly or using a stochastic approximation of the log-determinant's gradient, but both approaches remain computationally expensive for high-dimensional data. Our approach is the first injective model to provide a learnable manifold with exact and efficient log-determinant computation.

**Conformal networks.**   Numerous past works have imposed approximate conformality or its special cases as a regularizer [4, 31, 56, 57, 70], but it has been less common to enforce conformality strictly. To maintain orthogonal weights, one must optimize along the *Stiefel manifold* of orthogonal matrices. Past work to achieve this has either trained with Riemannian gradient descent or directly parameterized subsets of orthogonal matrices. Riemannian gradient descent algorithms typically require a singular value or QR decomposition at each training step [26, 30, 53]. We found that orthogonal matrices trained more quickly when directly parameterized. In particular, Lezcano-Casado and Martínez-Rubio [45] and Lezcano-Casado [44] parameterize orthogonal matrices as the matrix exponential of a skew-symmetric matrix, and Tomczak and Welling [67] use Householder matrices. We used a mix of both.

## 5   Experiments

To implement CEFs, we worked off of the nflows github repo [20], which is derived from the code of Durkan et al. [19]. Our code is available at `https://github.com/layer6ai-labs/CEF`. Full model and training details are provided in App. C, while additional reconstructions and generated images are presented in App. D.

### 5.1   Spherical Data

To demonstrate how a CEF can jointly learn a manifold and density, we generated a synthetic dataset from a known distribution with support on a spherical surface embedded in $\mathbb{R}^3$ as described in App. C. The distribution is visualized in Fig. 2, along with the training dataset of $10^3$ sampled points.

We trained the two components of the CEF jointly, using the mixed loss function in Eq. (7) with an end-to-end log-likelihood term. The resulting model density is plotted in Fig. 2 along with generated samples. It shows good fidelity to the true manifold and density.

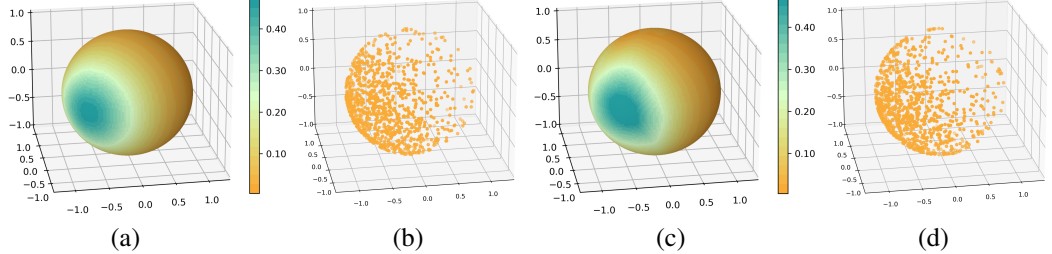

Figure 2: (a) A density $p_{\mathbf{x}}^*(\mathbf{x})$ with support on the sphere, and (b) $10^3$ samples comprising the training dataset $\{\mathbf{x_i}\}$. (c) The density learned by a CEF, and (d) $10^3$ generated samples.

### 5.2   Image Data

We now evaluate manifold flows on image data. Our aim is to show that, although they represent a strict architectural subset of mainstream injective flows, CEFs remain competitive in generative performance [6, 39]. In doing so, this work is the first to include end-to-end maximum likelihood

Table 3: Synthetic CIFAR-10 Ship Manifolds

| MODEL | 64 DIMENSIONS | | | | 512 DIMENSIONS | | | |
|---|---|---|---|---|---|---|---|---|
| | RECON | FID | DENSITY | COV | RECON | FID | DENSITY | COV |
| J-CEF | 0.000695 | 36.5 | 0.0491 | 0.0658 | 0.000568 | 76.2 | 0.398 | 0.266 |
| S-CEF | 0.000717 | 35.3 | 0.0548 | 0.0640 | 0.000627 | 74.7 | 0.421 | 0.251 |
| S-MF | 0.000469 | 28.7 | 0.0756 | 0.1103 | 0.000568 | 53.6 | 0.570 | 0.446 |

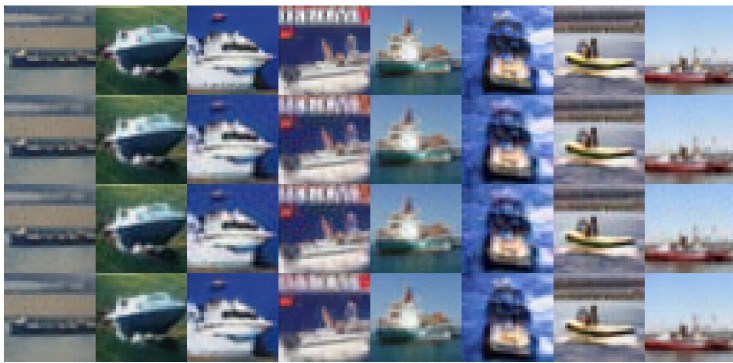

Figure 3: Synthetic 512-dimensional Ship Manifold Reconstructions. From top to bottom: groundtruth samples, joint CEF, sequential CEF, and sequential MF.

training with an injective flow on image data. Three approaches were evaluated on each dataset: a jointly trained CEF, a sequentially trained CEF, and for a baseline a sequentially trained injective flow, as in Brehmer and Cranmer [6], labelled *manifold flow* (MF).

Injective models cannot be compared on the basis of log-likelihood, since each model may have a different manifold support. Instead, we evaluate generative performance in terms of fidelity and diversity [61]. The *FID score* [28] is a single metric which combines these factors, whereas *density* and *coverage* [49] measure them separately. For FID, lower is better, while for density and coverage, higher is better. We use the PyTorch-FID package [62] and the implementation of density and coverage from Naeem et al. [49].

**Synthetic image manifolds.** Before graduating to natural high-dimensional data, we test CEFs on a high-dimensional synthetic data manifold whose properties are better understood. We generate data using a GAN pretrained on CIFAR-10 [40] by sampling from a selected number of latent space dimensions (64 and 512) with others held fixed. Specifically, we sample a single class from the class-conditional StyleGAN2-ADA provided by Karras et al. [33]. This setup reflects our model design in that (1) the true latent dimension is known and (2) since a single class is used, the resulting manifold is more likely to be connected. On the other hand, the GAN may not be completely injective, so its support may not technically be a manifold. Results are shown in Table 3, Figs. 3 and 4, and App. D.

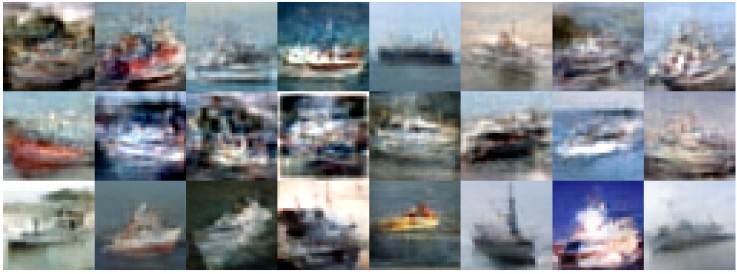

Figure 4: Uncurated Synthetic 512-dimensional Ship Manifold Samples. From top to bottom: joint CEF, sequential CEF, and sequential MF.

Table 4: Natural Image Data

| MODEL | MNIST | | | | CELEBA | | | |
|---|---|---|---|---|---|---|---|---|
| | RECON | FID | DENSITY | COV | RECON | FID | DENSITY | COV |
| J-CEF | 0.003222 | 38.5 | 0.0725 | 0.1796 | 0.001016 | 118 | 0.05581 | 0.00872 |
| S-CEF | 0.003315 | 37.9 | 0.0763 | 0.1800 | 0.001019 | 171 | 0.00922 | 0.00356 |
| S-MF | 0.000491 | 16.1 | 0.5003 | 0.7126 | 0.000547 | 142 | 0.02425 | 0.00576 |

Figure 5: MNIST Reconstructions. From top to bottom: groundtruth samples, joint CEF, sequential CEF, and sequential MF.

All models achieve comparable reconstruction losses with very minor visible artifacts, showing that conformal embeddings can learn complex manifolds with similar efficacy to state-of-the-art flows, despite their restricted architecture. The manifold flow achieves better generative performance based on our metrics and visual clarity. Between the CEFs, joint training allows the learned manifold to adapt better, but this did not translate directly to better generative performance.

**Natural image data.** We scale CEFs to natural data by training on the MNIST [42] and CelebA [46] datasets, for which a low-dimensional manifold structure is postulated but unknown. Results are given in Table 4, Figs. 5 and 6, and App. D.

As expected, since the MF's embedding is more flexible, it achieves smaller reconstruction losses than the CEFs. On MNIST, this is visible as faint blurriness in the CEF reconstructions in Fig. 5, and it translates to better sample quality for the MF as per the metrics in Table 4. Interestingly however, the jointly-trained CEF obtains substantially better sample quality on CelebA, both visually (Fig. 6) and by every metric. We posit this is due to the phenomenon observed by Caterini et al. [9] in concurrent work: for complex distributions, the learned manifold parameterization has significant influence on the difficulty of the density estimation task. Only the joint training approach, which maximizes likelihoods end-to-end, can train the manifold parameterization to an optimal starting point for density estimation, while sequential training optimizes the manifold solely on the basis of reconstruction loss. CelebA is the highest-dimensional dataset tested here, and its distribution is presumably quite complex, so one can reasonably expect joint training to provide better results. On the other hand, the sequentially trained CEF's performance suffers from the lack of both joint training and the expressivity afforded by the more general MF architecture.

## 6 Limitations and Future Directions

**Expressivity.** Just as standard flows trade expressivity for tractable likelihoods, so must injective flows. Our conformal embeddings in particular are less expressive than state-of-the-art flow models; they had higher reconstruction loss than the neural spline flow-based embeddings we tested. The conformal embeddings we designed were limited in that they mostly derive from dimension-*preserving* conformal mappings, which is a naturally restrictive class by Liouville's theorem [27]. Just as early work on NFs [16, 59] introduced limited classes of parameterizable bijections, which were later improved substantially (e.g. [19, 35]), our work introduces several classes of parameterizable conformal embeddings. We expect that future work will uncover more expressive conformal embeddings.

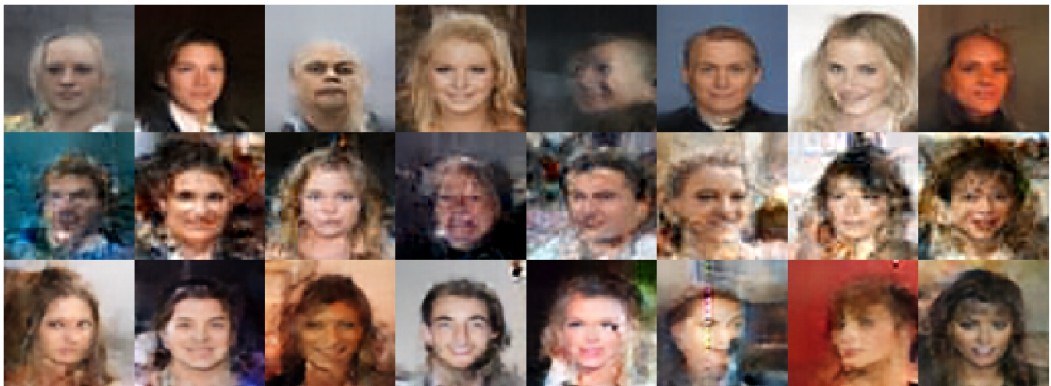

Figure 6: Uncurated CelebA Samples. From top to bottom: joint CEF, sequential CEF, and sequential MF.

**Manifold learning.** Strictly manifold-supported probability models such as ours introduce a bi-objective optimization problem. How to balance these objectives is unclear and, thus far, empirical [6]. The difference in supports between two manifold models also makes their likelihoods incomparable. Cunningham et al. [13] have made progress in this direction by convolving the manifold-supported distribution with noise, but this makes inference stochastic and introduces density estimation challenges. We suspect that using conformal manifold-learners may make density estimation more tractable in this setting, but further research is needed in this direction.

**Broader impact.** As deep generative models become more advanced, researchers should carefully consider some accompanying ethical concerns. Large-scale, natural image datasets carry social biases which are likely to be codified in turn by the models trained on them [65]. For instance, CelebA does not accurately represent the real-world distribution of human traits, and models trained on CelebA should be vetted for fairness before being deployed to make decisions that can adversely affect people. Deep generative modelling also lends itself to malicious practices [8] such as disinformation and impersonation using deepfakes [69].

Our work seeks to endow normalizing flows with more realistic assumptions about the data they model. While such improvements may invite malicious downstream applications, they also encode a better understanding of the data, which makes the model more interpretable and thus more transparent. We hope that a better understanding of deep generative models will synergize with current lines of research aimed at applying them for fair and explainable real-world use [3, 51].

## 7   Conclusion

This paper introduced Conformal Embedding Flows for modelling probability distributions on low-dimensional manifolds while maintaining tractable densities. We showed that conformal embeddings naturally match the framework of normalizing flows by providing efficient sampling, inference, and density estimation, and they are composable so that they can be scaled to depth. Furthermore, it appears conformality is a minimal restriction in that any looser condition will sacrifice one or more of these properties. As we have reviewed, previous instantiations of injective flows have not maintained all of these properties simultaneously.

Normalizing flows are still outperformed by other generative models such as GANs and VAEs in the arena of realistic image generation. Notably, these two alternatives benefit from a low-dimensional latent space, which better reflects image data's manifold structure and provides for more scalable model design. By equipping flows with a low-dimensional latent space, injective flow research has made progress towards VAE- or GAN-level performance. The CEF paradigm is a way to match these strides while maintaining the theoretical strengths of NFs.

## Acknowledgments and Disclosure of Funding

We thank Gabriel Loaiza-Ganem and Anthony Caterini for their valuable discussions and advice. We also thank Parsa Torabian for sharing his experience with orthogonal weights and Maksims Volkovs for his helpful feedback.

The authors declare no competing interests or third-party funding sources.

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
