# A   Injective Flows are Manifold Learners

## A.1   Densities on Manifolds

In this work we model a probability density $p$ on a Riemannian manifold $\mathcal{M}$. Here, we briefly review what this means formally.

Consider a probability measure $\mathbb{P}$ on the space $\mathcal{X} = \mathbb{R}^n$. We say $\mathbb{P}$ admits a density with respect to a base measure $\mu$ if $\mathbb{P}$ is absolutely continuous with respect to $\mu$. If so, we let the density be $p := \frac{d\mathbb{P}}{d\mu}$ (the Radon-Nikodym derivative of $\mathbb{P}$ with respect to $\mu$). The base measure $\mu$ is most commonly the Lebesgue measure on $\mathcal{X}$.

However, if $\mathbb{P}$ is supported on an $m$-dimensional Riemannian submanifold $\mathcal{M} \subset \mathcal{X}$, some adjustment is required. $\mathbb{P}$ will not be absolutely continuous with respect to the Lebesgue measure on $\mathcal{X}$, so we require a different choice of base measure. In the literature involving densities on manifolds (see Sec. 4), it is always assumed, but seldom explicitly stated, that the density's base measure is the Riemannian measure $\mu$ [55] of the submanifold $\mathcal{M}$. Furthermore, the Riemmanian metric from which $\mu$ arises is always inherited from the Euclidean metric of ambient space $\mathcal{X}$. From this construction we gather that, whenever a Riemannian submanifold $\mathcal{M} \subset \mathcal{X}$ is specified, a unique natural base measure $\mu$ follows.

Given an injective flow model $\mathbf{f}_\theta : \mathcal{Z} \to \mathcal{X}$ with latent space $\mathcal{Z} = \mathbb{R}^m$, the goal is to make the implied model manifold $\mathcal{M}_\theta = \mathbf{f}_\theta(\mathcal{Z})$ match the data manifold $\mathcal{M}_d$. As per the previous paragraph, this task equates to learning the base measure on which the model's density will be evaluated, which is necessarily a separate objective from likelihood maximization. Below we discuss how minimizing reconstruction loss achieves the goal of manifold matching.

## A.2   Reconstruction Minimization for Manifold Learning

Let $\mathbf{f}_\theta : \mathcal{Z} \to \mathcal{X} = \mathbb{R}^n$ be an injective flow ($m < n$) with a smooth left-inverse $\mathbf{f}_\theta^\dagger$ as described in Sec. 2.2. By construction, the image $\mathcal{M}_\theta = \mathbf{f}_\theta(\mathcal{Z})$ of the flow is a Riemannian submanifold of $\mathcal{X}$; we call $\mathcal{M}_\theta$ the *model manifold*. Let $P$ be a data distribution supported by an $m$-dimensional Riemannian submanifold $\mathcal{M}_d$; we call this the *data manifold*. Suppose furthermore that $P$ admits a probability density with respect to the Riemannian measure of $\mathcal{M}_d$.

**Proposition 1**   $\mathcal{R}(\theta) := \mathbb{E}_{\mathbf{x} \sim P} ||\mathbf{x} - \mathbf{f}_\theta(\mathbf{f}_\theta^\dagger(\mathbf{x}))||^2 = 0$ *if and only if* $\mathcal{M}_d \subseteq cl\,(\mathcal{M}_\theta)$.

Since the reconstruction loss $\mathcal{R}(\theta)$ is continuous, we infer that $\mathcal{R}(\theta) \to 0$ will bring $\mathcal{M}_\theta$ and $\mathcal{M}_d$ into alignment (except possibly for the $P$-null set $\mathcal{M}_d \cap \partial\mathcal{M}_\theta$).

**Proof**   For the forward direction, suppose $\mathcal{R}(\theta) = 0$. For a single point $\mathbf{x} \in \mathbb{R}^n$, by the definition of $\mathbf{f}_\theta^\dagger$ we have $\mathbf{x} = \mathbf{f}_\theta(\mathbf{f}_\theta^\dagger(\mathbf{x}))$ if and only if $\mathbf{x} \in \mathbf{f}_\theta(\mathcal{Z}) = \mathcal{M}_\theta$. Put differently, $\mathcal{M}_\theta = \{\mathbf{x} \in \mathcal{X} : \mathbf{x} = \mathbf{f}_\theta(\mathbf{f}_\theta^\dagger(\mathbf{x}))\}$, so a reconstruction error of zero implies $P(\mathcal{M}_\theta) = 1$. This means that $P$'s support must by definition be a subset of $cl\,(\mathcal{M}_\theta)$. It follows that $\mathcal{M}_d \subseteq cl\,(\mathcal{M}_\theta)$.

For the reverse direction, note that if $\mathcal{M}_d \subseteq cl\,(\mathcal{M}_\theta)$, then $\mathcal{M}_d \setminus \mathbf{f}_\theta(\mathcal{Z})$ has measure zero in $\mathcal{M}_d$, so $\mathbf{f}_\theta(\mathbf{f}_\theta^\dagger(\mathbf{x})) = 0$ $P$-almost surely. This fact yields $\mathcal{R}(\theta) = 0$. $\square$

## A.3   Joint Training and Wasserstein Training

The Wasserstein-1 distance between the groundtruth and model distributions is another objective motivated by the low-dimensional manifold structure of high-dimensional data. In some sense, minimizing Wasserstein-1 distance is a more elegant approach than sequential or joint training because it is a distance metric between probability distributions, whereas the sequential and joint objectives involve separate terms for the support and the likelihood within. However, in practice the Wasserstein distance cannot be estimated without bias in a polynomial number of samples [2], and it must be estimated adversarially [1, 66].

We compare our joint training method to Wasserstein training on 64000 points sampled from a 2D Gaussian mixture embedded as a plane in 3D space. The conformal embedding $\mathbf{g}$ is a simple orthogonal transformation from 2 into 3 dimensions, which makes the manifold easy to plot. We let

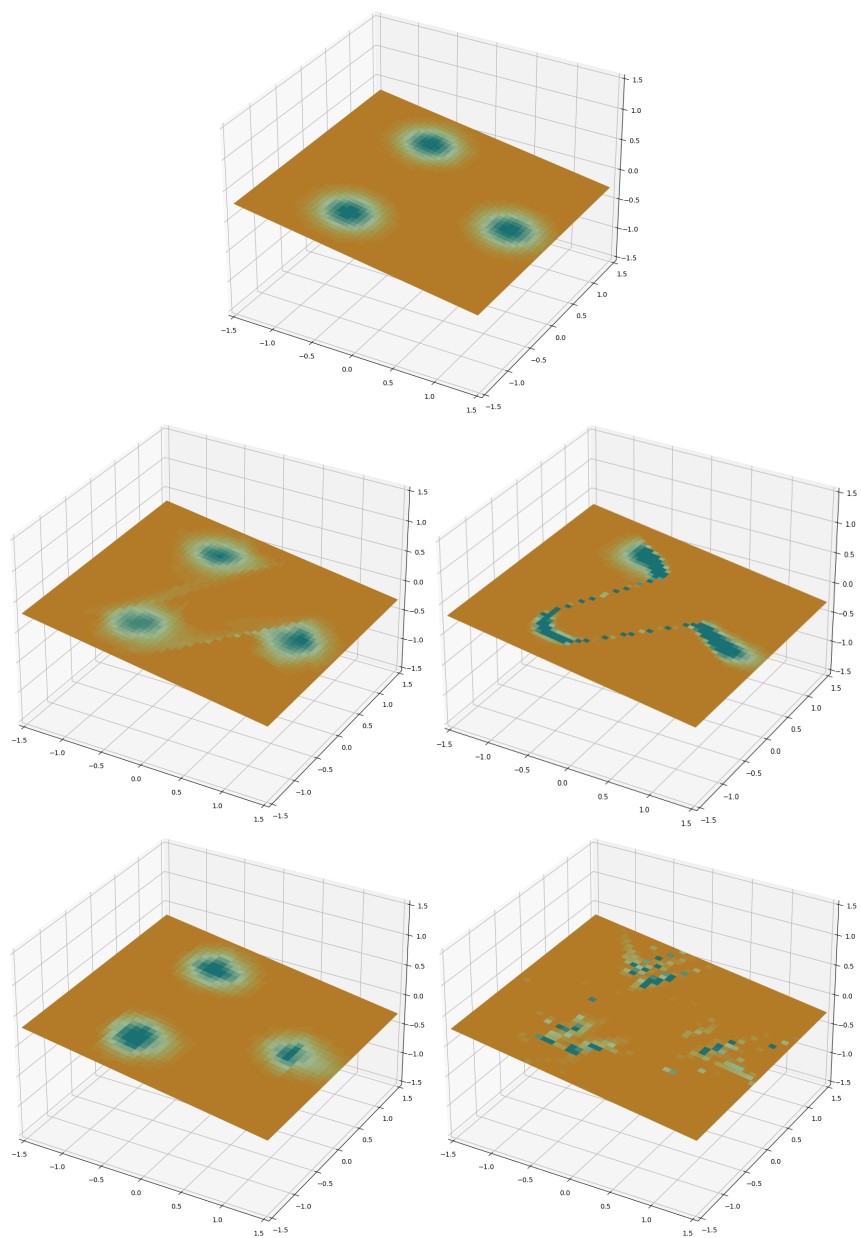

Figure 7: Gaussian mixture on a 2D plane. From top to bottom: groundtruth distribution, the learned distribution using affine coupling layers, and learned distribution using rational-quadratic coupling layers. The results of joint training are shown on the left, while the results of Wasserstein adversarial training are shown on the right.

**h** be a simple flow consisting of 3 LU-decomposed linear transformations interspersed with coupling layers, where each layer has 2 residual blocks with 128 hidden features. Both affine coupling and rational-quadratic coupling layers were tested.

Each architecture was separately trained with both the joint loss and with adversarially estimated Wasserstein loss. The discriminator was a 16-hidden layer ReLU MLP with 512 hidden units each. We enforce the Lipschitz constraint using gradient penalties [25]. Results are visible in Fig. 7.

When adversarially trained, the inductive bias of the generator appears to play a strong role, and neither model learns the density well. These results suggest that training a flow jointly provides better density estimation than estimating the Wasserstein loss. Our observations support those of Grover

et al. [24], who show that adversarial training can be counterproductive to likelihood maximization, and Brehmer and Cranmer [6], who report poor results when their model is trained for optimal transport.

# B  Details on Conformal Embeddings and Conformal Mappings

Let $(\mathcal{U}, \eta_{\mathbf{u}})$ and $(\mathcal{X}, \eta_{\mathbf{x}})$ be two Riemannian manifolds. We define a diffeomorphism $\mathbf{f} : \mathcal{U} \to \mathcal{X}$ to be a *conformal diffeomorphism* if it pulls back the metric $\eta_{\mathbf{x}}$ to be some non-zero scalar multiple of $\eta_{\mathbf{u}}$ [43]. That is,

$$\mathbf{f}^* \eta_{\mathbf{x}} = \lambda^2 \eta_{\mathbf{u}} \tag{10}$$

for some smooth non-zero scalar function $\lambda$ on $\mathcal{U}$. Furthermore, we define a smooth embedding $\mathbf{g} : \mathcal{U} \to \mathcal{X}$ to be a *conformal embedding* if it is a conformal diffeomorphism onto its image $(\mathbf{g}(\mathcal{U}), \eta_{\mathbf{x}})$, where $\eta_{\mathbf{x}}$ is inherited from the ambient space $\mathcal{X}$.

In our context, $\mathcal{U} \subseteq \mathbb{R}^m$, $\mathcal{X} = \mathbb{R}^n$, and $\eta_{\mathbf{u}}$ and $\eta_{\mathbf{x}}$ are Euclidean metrics. This leads to an equivalent property (Eq. (5)):

$$\mathbf{J}_{\mathbf{g}}^T(\mathbf{u}) \mathbf{J}_{\mathbf{g}}(\mathbf{u}) = \lambda^2(\mathbf{u}) \mathbf{I}_m. \tag{11}$$

This also guarantees that $\det[\mathbf{J}_{\mathbf{g}}^T \mathbf{J}_{\mathbf{g}}] = \lambda^{2m}$ is tractable, even when $\mathbf{g} = \mathbf{g}_k \circ ... \circ \mathbf{g}_1$ is composed from several layers, as is needed for scalable injective flows.

To demonstrate that conformal embeddings are an expressive class of functions, we first turn to the most restricted case where $n = m$; i.e. conformal mappings. In Apps. B.1 and B.2 we provide an intuitive investigation of the classes of conformal mappings using infinitesimals. We then discuss in App. B.3 why conformal embeddings in general are more challenging to analyze, but also show intuitively why they are more expressive than dimension-preserving conformal mappings.

## B.1  Infinitesimal Conformal Mappings

Consider a mapping of Euclidean space with dimension $m \geq 3$. Liouville's theorem for conformal mappings constrains the set of such maps which satisfy the conformal condition Eq. (11). Such functions can be decomposed into translations, orthogonal transformations, scalings, and inversions. Here we provide a direct approach for the interested reader, which also leads to some insight on the general case of conformal embeddings [15]. First we will find all infinitesimal transformations which satisfy the conformal condition, then exponentiate them to obtain the set of finite conformal mappings.

Consider a transformation $\mathbf{f} : \mathbb{R}^m \to \mathbb{R}^m$ which is infinitesimally close to the identity function, expressed in Cartesian coordinates as

$$\mathbf{f}(\mathbf{x}) = \mathbf{x} + \boldsymbol{\epsilon}(\mathbf{x}). \tag{12}$$

That is, we only keep terms linear in the infinitesimal quantity $\boldsymbol{\epsilon}$. The mappings produced will only encompass transformations which are continuously connected to the identity, but we restrict our attention to these for now. However, this simple form allows us to directly study how Eq. (11) constrains the infinitesimal $\boldsymbol{\epsilon}(\mathbf{x})$:

$$\begin{aligned} \mathbf{J}_{\mathbf{f}}^T(\mathbf{x}) \mathbf{J}_{\mathbf{f}}(\mathbf{x}) &= \left[ \mathbf{I}_m + \frac{\partial \boldsymbol{\epsilon}}{\partial \mathbf{x}} \right]^T \left[ \mathbf{I}_m + \frac{\partial \boldsymbol{\epsilon}}{\partial \mathbf{x}} \right] \\ &= \mathbf{I}_m + \frac{\partial \boldsymbol{\epsilon}}{\partial \mathbf{x}}^T + \frac{\partial \boldsymbol{\epsilon}}{\partial \mathbf{x}}. \end{aligned} \tag{13}$$

By Eq. (11), the symmetric sum of $\partial \boldsymbol{\epsilon} / \partial \mathbf{x}$ must be proportional to the identity matrix. Let us call the position-dependent proportionality factor $\eta(\mathbf{x})$. We can start to understand $\eta(\mathbf{x})$ by taking a trace

$$\frac{\partial \boldsymbol{\epsilon}}{\partial \mathbf{x}}^T + \frac{\partial \boldsymbol{\epsilon}}{\partial \mathbf{x}} = \eta(\mathbf{x}) \mathbf{I}_m, \tag{14}$$

$$\frac{2}{m} \text{tr} \left( \frac{\partial \boldsymbol{\epsilon}}{\partial \mathbf{x}} \right) = \eta(\mathbf{x}). \tag{15}$$

Taking another derivative of Eq. (14) proves to be useful, so we switch to index notation to handle the tensor multiplications,

$$\frac{\partial}{\partial x_k}\frac{\partial \epsilon_j}{\partial x_i} + \frac{\partial}{\partial x_k}\frac{\partial \epsilon_i}{\partial x_j} = \frac{\partial \eta}{\partial x_k}\delta_{ij}, \tag{16}$$

where the Kronecker delta $\delta_{ij}$ is 1 if $i = j$, and 0 otherwise. On the left-hand-side, derivatives can be commuted. By taking a linear combination of the three permutations of indices we come to

$$2\frac{\partial}{\partial x_k}\frac{\partial \epsilon_i}{\partial x_j} = \frac{\partial \eta}{\partial x_j}\delta_{ik} + \frac{\partial \eta}{\partial x_k}\delta_{ij} - \frac{\partial \eta}{\partial x_i}\delta_{jk}. \tag{17}$$

Summing over elements where $j = k$ gives the Laplacian of $\epsilon_i$, while picking up only the derivatives of $\eta$ with respect to $x_i$, so we can switch back to vector notation where

$$2\nabla^2\boldsymbol{\epsilon} = (2 - m)\frac{\partial \eta}{\partial \mathbf{x}}. \tag{18}$$

Now we have two equations (14) and (18)[4] involving derivatives of $\boldsymbol{\epsilon}$ and $\eta$. To eliminate $\boldsymbol{\epsilon}$, we can apply $\nabla^2$ to (14), while applying $\partial/\partial \mathbf{x}$ to (18)

$$\nabla^2\frac{\partial \boldsymbol{\epsilon}}{\partial \mathbf{x}}^T + \nabla^2\frac{\partial \boldsymbol{\epsilon}}{\partial \mathbf{x}} = \nabla^2\eta\mathbf{I}_d \tag{19}$$

$$2\nabla^2\frac{\partial \boldsymbol{\epsilon}}{\partial \mathbf{x}} = (2 - m)\frac{\partial^2\eta}{\partial \mathbf{x}\partial \mathbf{x}}. \tag{20}$$

Since Eq. (20) is manifestly symmetric, the left-hand-sides are actually equal. Equating the right-hand-sides, we can again sum the diagonal terms, giving the much simpler form

$$(m - 1)\nabla^2\eta = 0. \tag{21}$$

Ultimately, revisiting Eq. (20) shows that the function $\eta(\mathbf{x})$ is linear in the coordinates

$$\frac{\partial^2\eta}{\partial \mathbf{x}\partial \mathbf{x}} = 0 \implies \eta(\mathbf{x}) = \alpha + \boldsymbol{\beta} \cdot \mathbf{x}, \tag{22}$$

for constants $\alpha, \boldsymbol{\beta}$. This allows us to relate back to the quantity of interest $\boldsymbol{\epsilon}$. Skimming back over the results so far, the most general equation where having the linear expression for $\eta(\mathbf{x})$ helps is Eq. (17) which now is

$$2\frac{\partial}{\partial x_k}\frac{\partial \epsilon_i}{\partial x_j} = \beta_j\delta_{ik} + \beta_k\delta_{ij} - \beta_i\delta_{jk}. \tag{23}$$

The point is that the right-hand-side is constant, meaning that $\boldsymbol{\epsilon}(\mathbf{x})$ is at most quadratic in $\mathbf{x}$. Hence, we can make an ansatz for $\boldsymbol{\epsilon}$ in full generality, involving sets of infinitesimal constants

$$\boldsymbol{\epsilon} = \mathbf{a} + \mathbf{B}\mathbf{x} + \mathbf{x}\overleftrightarrow{\mathbf{C}}\mathbf{x}, \tag{24}$$

where $\overleftrightarrow{\mathbf{C}} \in \mathbb{R}^{m\times m\times m}$ is a 3-tensor.

So far we have found that infinitesimal conformal transformations can have at most quadratic dependence on the coordinates. It remains to determine the constraints on each set of constants $\mathbf{a}$, $\mathbf{B}$, and $\overleftrightarrow{\mathbf{C}}$, and interpret the corresponding mappings. We consider each of them in turn.

All constraints on $\boldsymbol{\epsilon}$ involve derivatives, so there is nothing more to say about the constant term. It represents an infinitesimal translation

$$\mathbf{f}(\mathbf{x}) = \mathbf{x} + \mathbf{a}. \tag{25}$$

On the other hand, the linear term is constrained by Eqs. (14) and (15) which give

$$\mathbf{B} + \mathbf{B}^T = \frac{2}{m}\text{tr}(\mathbf{B})\mathbf{I}_m. \tag{26}$$

---

[4]We note that the steps following Eq. (18) are only justified for $m \geq 3$ which we have assumed. In two dimensions the conformal group is much larger and Liouville's theorem no longer captures all conformal mappings.

Hence, $\mathbf{B}$ has an unconstrained anti-symmetric part $\mathbf{B}_{\text{AS}} = \frac{1}{2}(\mathbf{B} - \mathbf{B}^T)$ representing an infinitesimal rotation

$$\mathbf{f}(\mathbf{x}) = \mathbf{x} + \mathbf{B}_{\text{AS}}\mathbf{x}, \tag{27}$$

while its symmetric part is diagonal as in Eq. (26),

$$\mathbf{f}(\mathbf{x}) = \mathbf{x} + \lambda\mathbf{x}, \quad \lambda = \frac{1}{m}\text{tr}(\mathbf{B}), \tag{28}$$

which is an infinitesimal scaling. This leaves only the quadratic term for interpretation which is more easily handled in index notation, i.e. $\epsilon_i = \sum_{lm} C_{ilm}x_l x_m$. The quadratic term is significantly restricted by Eq. (23),

$$2\frac{\partial^2}{\partial x_k \partial x_j} \sum_{lm} C_{ilm}x_l x_m = 2C_{ijk} = \beta_j \delta_{ik} + \beta_k \delta_{ij} - \beta_i \delta_{jk}. \tag{29}$$

This allows us to isolate $\beta_k$ in terms of $C_{ijk}$, specifically from the trace over $C$'s first two indices,

$$2\sum_{i=j} C_{ijk} = \beta_k + \beta_k m - \beta_k = \beta_k m. \tag{30}$$

Hereafter we use $b_k = \beta_k/2 = \sum_{i=j} C_{ijk}/m$. Then with Eq. (29) the corresponding infinitesimal transformation is

$$\begin{aligned}
f_i(\mathbf{x}) &= x_i + \sum_{jk} C_{ijk}x_j x_k \\
&= x_i + \sum_{jk} (b_j \delta_{ik} + b_k \delta_{ij} - b_i \delta_{jk})x_j x_k \\
&= x_i + 2x_i \sum_j b_j x_j - b_i \sum_j (x_j)^2,
\end{aligned} \tag{31}$$

$$\mathbf{f}(\mathbf{x}) = \mathbf{x} + 2(\mathbf{b} \cdot \mathbf{x})\mathbf{x} - \|\mathbf{x}\|^2 \mathbf{b}.$$

We postpone the interpretation momentarily.

Thus we have found all continuously parametrizable infinitesimal conformal mappings connected to the identity and showed they come in four distinct types. By composing infinitely many such transformations, or "exponentiating" them, we obtain finite conformal mappings. Formally, this is the process of exponentiating the elements of a Lie algebra to obtain elements of a corresponding Lie group.

## B.2 Finite Conformal Mappings

As an example of obtaining finite mappings from infinitesimal ones we take the infinitesimal rotations from Eq. (27) where we note that $\mathbf{f}$ only deviates from the identity by an infinitesimal vector field $\mathbf{B}_{\text{AS}}\mathbf{x}$. By integrating the field we get the finite displacement of any point under many applications of $\mathbf{f}$, i.e. the integral curves $\mathbf{x}(t)$ defined by

$$\dot{\mathbf{x}}(t) = \mathbf{B}_{\text{AS}}\mathbf{x}(t), \quad \mathbf{x}(0) = \mathbf{x}_0. \tag{32}$$

This differential equation has the simple solution

$$\mathbf{x}(t) = \exp(t\mathbf{B}_{\text{AS}})\mathbf{x}_0. \tag{33}$$

Finally we recognize that when a matrix $\mathbf{A}$ is antisymmetric, the matrix exponential $e^{\mathbf{A}}$ is orthogonal, showing that the finite transformation given by $t = 1$, $\mathbf{f}(\mathbf{x}_0) = \exp(\mathbf{B}_{\text{AS}})\mathbf{x}_0$, is indeed a rotation. Furthermore, it is intuitive that infinitesimal translations and scalings also compose into finite translations and scalings. Examples are shown in Fig. 8 (a-c)

The infinitesimal transformation in Eq. (31) is non-linear in $\mathbf{x}$, so it does not exponentiate easily as for the other three cases. It helps to linearize with a change of coordinates $\mathbf{y} = \mathbf{x}/\|\mathbf{x}\|^2$ which happens to be an inversion:

$$\dot{\mathbf{x}}(t) = 2(\mathbf{b} \cdot \mathbf{x})\mathbf{x} - \|\mathbf{x}\|^2 \mathbf{b}, \tag{34}$$

$$\dot{\mathbf{y}}(t) = \frac{\dot{\mathbf{x}}}{\|\mathbf{x}\|^2} - 2\frac{\mathbf{x} \cdot \dot{\mathbf{x}}}{\|\mathbf{x}\|^4}\mathbf{x} = -\mathbf{b}. \tag{35}$$

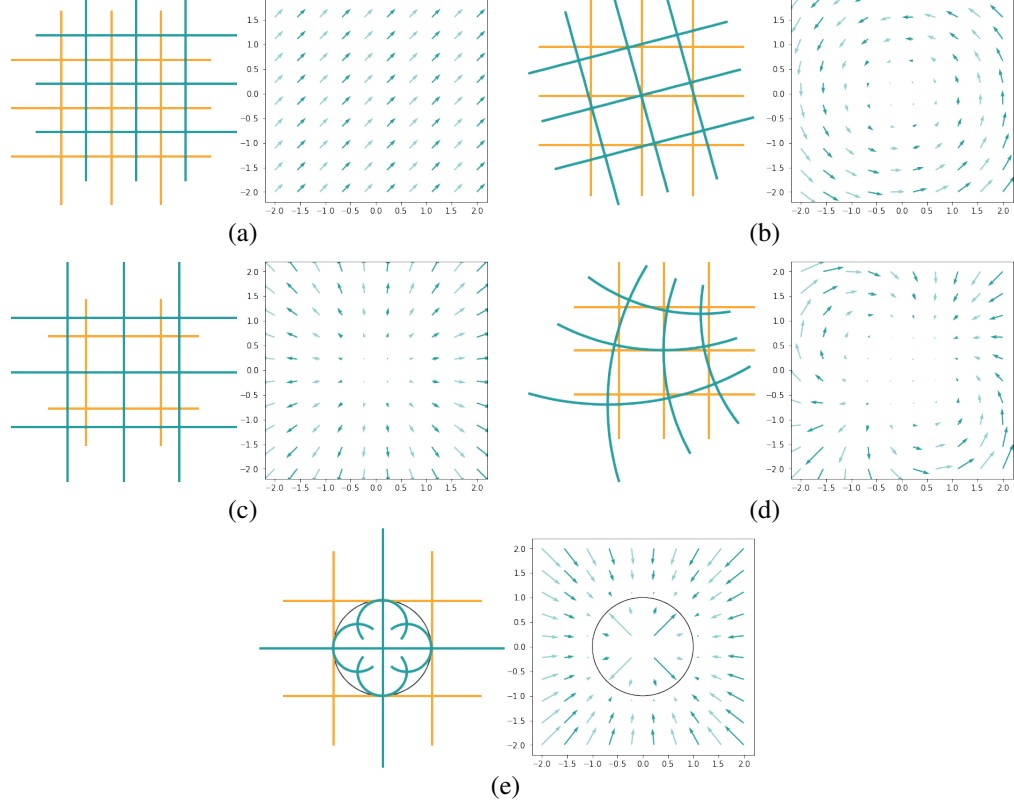

Figure 8: Effects of conformal mappings on gridlines, and their corresponding vector fields showing local displacements. Mappings are: (a) translation by $\mathbf{a} = [1, 1]$; (b) orthogonal transformation (2D rotation) by angle $\theta = \pi/12$; (c) scaling by $\lambda = 1.5$; (d) SCT by $\mathbf{b} = [-0.1, -0.1]$; (e) inversion, also showing the unit circle. The interior of the circle is mapped to the exterior, and vice versa.

We now get the incredibly simple solution $\mathbf{y}(t) = \mathbf{y}_0 - t\mathbf{b}$, a translation, after which we can undo the inversion

$$\frac{\mathbf{x}(t)}{\|\mathbf{x}\|^2} = \frac{\mathbf{x}_0}{\|\mathbf{x}_0\|^2} - t\mathbf{b}. \tag{36}$$

This form is equivalent to a Special Conformal Transformation (SCT) [15], which we can see by defining the finite transformation as $\mathbf{f}(\mathbf{x}_0) = \mathbf{x}(1)$, and taking the inner product of both sides with themselves

$$\|\mathbf{f}(\mathbf{x}_0)\|^2 = \frac{\|\mathbf{x}_0\|^2}{1 - 2\mathbf{b} \cdot \mathbf{x}_0 + \|\mathbf{b}\|^2\|\mathbf{x}_0\|^2}, \tag{37}$$

and finally isolating

$$\mathbf{f}(\mathbf{x}_0) = \frac{\|\mathbf{f}(\mathbf{x}_0)\|^2}{\|\mathbf{x}_0\|^2}\mathbf{x}_0 - \|\mathbf{f}(\mathbf{x}_0)\|^2\mathbf{b} = \frac{\mathbf{x}_0 - \|\mathbf{x}_0\|^2\mathbf{b}}{1 - 2\mathbf{b} \cdot \mathbf{x}_0 + \|\mathbf{b}\|^2\|\mathbf{x}_0\|^2}. \tag{38}$$

An example SCT is shown in Fig. 8 (d), demonstrating their non-linear nature. In the process of this derivation we have learned that SCTs can be interpreted as an inversion, followed by a translation by $-\mathbf{b}$, followed by an inversion, and the infinitesimal Eq. (31) is recovered when the translation is small.

By composition, the four types of finite conformal mapping we have encountered, namely translations, rotations, scalings, and SCTs, generate the conformal group - the group of transformations of Euclidean space which locally preserve angles and orientation. The infinitesimal transformations we derived directly give the corresponding elements of the Lie algebra.

Eq. (11) also admits non-orientation preserving solutions which are not generated by the infinitesimal approach. Composing the scalings in Eq. (28) only produces finite scalings by a positive factor,

i.e. $\mathbf{f}(\mathbf{x}) = e^\lambda \mathbf{x}$. Similarly, composing infinitesimal rotations does not generate reflections - non-orientation preserving orthogonal transformations that are not connected to the identity. The conformal group can be extended by including non-orientation preserving transformations, namely inversions (Fig. 8 (e)), negative scalings, and reflections as in Table 1. All of these elements still satisfy Eq. (11), as do their closure under composition. By Liouville's theorem, these comprise all possible conformal mappings.

The important point for our discussion is that any conformal mapping can be built up from the simple elements in Table 1. In other words, a neural network can learn any conformal mapping by representing a sequence of the simple elements.

### B.3 Conformal Embeddings

Whereas conformal mappings have been exhaustively classified, conformal embeddings have not. While the defining equations for a conformal embedding $\mathbf{g} : \mathcal{U} \to \mathcal{X}$, namely

$$\mathbf{J}_\mathbf{g}^T(\mathbf{u})\mathbf{J}_\mathbf{g}(\mathbf{u}) = \lambda^2(\mathbf{u})\mathbf{I}_m, \tag{39}$$

appear similar to those of conformal mappings, we cannot apply the techniques from Apps. B.1 and B.2 to enumerate them. Conformal embeddings do not necessarily have identical domain and codomain. As such, finite conformal embeddings can not be generated by exponentiating infinitesimals.

The lack of full characterization of conformal embeddings hints that they are a richer class of functions. For a more concrete understanding, we can study Eq. (39) as a system of PDEs. This system consists of $m(m+1)/2$ independent equations (noting the symmetry of $\mathbf{J}_\mathbf{g}^T\mathbf{J}_\mathbf{g}$) to be satisfied by $n+1$ functions, namely $\mathbf{g}(\mathbf{u})$ and $\lambda(\mathbf{u})$. In the typical case that $n < m(m+1)/2 - 1$, i.e. $n$ is not significantly larger than $m$, the system is overdetermined. Despite this, solutions do exist. We have already seen that the most restricted case $n = m$ of conformal mappings admits four qualitatively different classes of solutions. These remain solutions when $n > m$ simply by having $\mathbf{g}$ map to a constant in the extra $n - m$ dimensions.

Intuitively, adding an extra dimension for solving the PDEs is similar to introducing a slack variable in an optimization problem. In case it is not clear that adding additional functions $\mathbf{g}_i, i > m$ enlarges the class of solutions of Eq. (39), we provide a concrete example. Take the case $n = m = 2$ for a fixed $\lambda(u_1, u_2)$. The system of equations that $\mathbf{g}(\mathbf{u})$ must solve is

$$\left(\frac{\partial g_1}{\partial u_1}\right)^2 + \left(\frac{\partial g_2}{\partial u_1}\right)^2 = \lambda^2(u_1, u_2),$$

$$\left(\frac{\partial g_1}{\partial u_2}\right)^2 + \left(\frac{\partial g_2}{\partial u_2}\right)^2 = \lambda^2(u_1, u_2), \tag{40}$$

$$\frac{\partial g_1}{\partial u_1}\frac{\partial g_1}{\partial u_2} + \frac{\partial g_2}{\partial u_1}\frac{\partial g_2}{\partial u_2} = 0.$$

Suppose that for the given $\lambda(u_1, u_2)$ no complete solution exists, but we do have a $\mathbf{g}(\mathbf{u})$ which simultaneously solves all but the first equation. Enlarging the codomain $\mathcal{X}$ with an additional dimension ($n = 3$) gives an additional function $g_3(\mathbf{u})$ to work with while $\lambda(u_1, u_2)$ is unchanged. The system of equations becomes

$$\left(\frac{\partial g_1}{\partial u_1}\right)^2 + \left(\frac{\partial g_2}{\partial u_1}\right)^2 + \left(\frac{\partial g_3}{\partial u_1}\right)^2 = \lambda^2(u_1, u_2),$$

$$\left(\frac{\partial g_1}{\partial u_2}\right)^2 + \left(\frac{\partial g_2}{\partial u_2}\right)^2 + \left(\frac{\partial g_3}{\partial u_2}\right)^2 = \lambda^2(u_1, u_2), \tag{41}$$

$$\frac{\partial g_1}{\partial u_1}\frac{\partial g_1}{\partial u_2} + \frac{\partial g_2}{\partial u_1}\frac{\partial g_2}{\partial u_2} + \frac{\partial g_3}{\partial u_1}\frac{\partial g_3}{\partial u_2} = 0.$$

Our partial solution can be worked into an actual solution by letting $g_3$ satisfy

$$\left(\frac{\partial g_3}{\partial u_1}\right)^2 = \lambda^2(u_1, u_2) - \left(\frac{\partial g_1}{\partial u_1}\right)^2 - \left(\frac{\partial g_2}{\partial u_1}\right)^2, \tag{42}$$

with all other derivatives of $g_3$ vanishing. Hence $g_3$ is constant in all directions except the $u_1$ direction so that, geometrically speaking, the $u_1$ direction is bent and warped by the embedding into the additional $x_3$ dimension.

To summarize, compared to conformal mappings, with dimension-changing conformal embeddings the number of equations in the system remains the same but the number of functions available to satisfy them increases. This allows conformal embeddings to be much more expressive than the fixed set of conformal mappings, but also prevents an explicit classification and parametrization of all conformal embeddings.

## C Experimental Details

Table 5: Network parameters for each embedding $\mathbf{g}$ and low-dimensional flow $\mathbf{h}$

| METHOD | DATASET | | | |
|---|---|---|---|---|
| | SHIP $m = 64$ | SHIP $m = 512$ | MNIST | CELEBA |
| CEF $\mathbf{g}$ | 270,918 | 1,647,174 | 139,460 | 23,649 |
| MF $\mathbf{g}$ | 2,276,508 | 2,276,508 | 3,135,428 | 2,311,212 |
| $\mathbf{h}$ | 16,978,432 | 49,381,376 | 21,410,816 | 418,136,600 |

### C.1 Synthetic Spherical Distribution

**Model.** The conformal embedding $\mathbf{g}$ was composed of a padding layer, SCT, orthogonal transformation, translation, and scaling (see App. B.2 for the definition of SCT). The base flow $\mathbf{h}$ used two coupling layers backed by rational quadratic splines with 16 hidden units.

**Training.** The CEF components were trained jointly on the mixed loss function in Eq. (7) with an end-to-end log-likelihood term for 45 epochs. The reconstruction loss had weight 10000, and the log-likelihood had weight 10. We used a batch size of 100 and a learning rate of $5 \times 10^{-3}$ with the Adam optimizer.

**Data.** For illustrative purposes we generated a synthetic dataset from a known distribution on a spherical surface embedded in $\mathbb{R}^3$. The sphere is a natural manifold with which to demonstrate learning a conformal embedding with a CEF, since we can analytically find suitable maps $\mathbf{g} : \mathbb{R}^2 \to \mathbb{R}^3$ that embed the sphere[5] with Cartesian coordinates describing both spaces. For instance consider

$$\mathbf{g} = \left( \frac{2r^2 z_1}{z_1^2 + z_2^2 + r^2}, \frac{2r^2 z_2}{z_1^2 + z_2^2 + r^2}, r \frac{z_1^2 + z_2^2 - r^2}{z_1^2 + z_2^2 + r^2} \right), \tag{43}$$

where $r \in \mathbb{R}$ is a parameter. Geometrically, this embedding takes the domain manifold, viewed as the surface $x_3 = 0$ in $\mathbb{R}^3$, and bends it into a sphere of radius $r$ centered at the origin. Computing the Jacobian directly gives

$$\mathbf{J}_{\mathbf{g}}^T \mathbf{J}_{\mathbf{g}} = \frac{4r^4}{(z_1^2 + z_2^2 + r^2)^2} \mathbf{I}_2, \tag{44}$$

which shows that $\mathbf{g}$ is a conformal embedding (Eq. (5)) with $\lambda(\mathbf{z}) = \frac{2r^2}{z_1^2 + z_2^2 + r^2}$. Of course, this $\mathbf{g}$ is also known as a *stereographic projection*, but here we view its codomain as all of $\mathbb{R}^3$, rather than the 2-sphere.

With this in mind it is not surprising that a CEF can learn an embedding of the sphere, but we would still like to study how a density confined to the sphere is learned. Starting with a multivariate Normal $\mathcal{N}(\boldsymbol{\mu}, \mathbf{I}_3)$ in three dimensions we drew samples and projected them radially onto the unit sphere. This yields the density given by integrating out the radial coordinate from the standard Normal distribution:

$$p_{\mathcal{M}}(\phi, \theta) = \int_0^\infty \frac{1}{(2\pi)^{3/2}} \exp\left\{ -\frac{1}{2}\left(r^2 - 2r\left(\cos\phi\sin\theta, \sin\phi\sin\theta, \cos\theta\right) \cdot \boldsymbol{\mu} + \|\boldsymbol{\mu}\|^2\right) \right\} r^2 dr. \tag{45}$$

---

[5]Technically the "north pole" of the sphere $(0, 0, 1)$ is not in the range of $\mathbf{g}$, which leaves a manifold $\mathbb{S}^2 \backslash \{\text{north pole}\}$ that is topologically equivalent to $\mathbb{R}^2$.

With the shorthand $\mathbf{t} = (\cos\phi\sin\theta,\ \sin\phi\sin\theta,\ \cos\theta)$ for the angular direction vector, the integration can be performed

$$p_{\mathcal{M}}(\phi,\theta) = \frac{1}{2^{5/2}\pi^{3/2}} e^{-\|\boldsymbol{\mu}\|^2/2} \Big( 2\mathbf{t}\cdot\boldsymbol{\mu} + \sqrt{2\pi}\left((\mathbf{t}\cdot\boldsymbol{\mu})^2 + 1\right) e^{(\mathbf{t}\cdot\boldsymbol{\mu})^2/2}\left(\mathrm{erf}\left(\mathbf{t}\cdot\boldsymbol{\mu}/\sqrt{2}\right) + 1\right) \Big). \tag{46}$$

This distribution is visualized in Fig. 2 for the parameter $\boldsymbol{\mu} = (-1,-1,0)$.

## C.2 Synthetic CIFAR-10 Ship Manifolds

**Dataset.** To generate the 64- and 512-dimensional synthetic datasets, we sample from *ship* class of the pretrained class-conditional StyleGAN2-ADA provided in PyTorch by Karras et al. [33]. To generate a sample of dimension $m$, we first randomly sample entries for all but $m$ latent dimensions, fix these, then repeatedly sample the remaining $m$ to generate the dataset. We use a training size of 20000 for $m = 64$ and 50000 when $m = 512$ of which we hold out a tenth of the data for validation when training. We generate an extra 10000 samples from each distribution for testing.

**Models.** All models for each dimension $m \in \{64, 512\}$ use the same architecture for their $\mathbf{h}$ components: a simple 8-layer rational-quadratic neural spline flow with 3 residual blocks per layer and 512 hidden channels each. It is applied to flattened data of dimension $m$.

The baseline's embedding $\mathbf{g}$ is a rational-quadratic neural spline flow network of 3 levels, 3 steps per level, and 3 residual blocks per step with 64 hidden channels each. The output of each scale is reshaped into $8 \times 8$, and the outputs of all scales are concatenated. We then apply an invertible $1 \times 1$ convolution, and project and flatten the input down to $m$ dimensions.

On the other hand, both CEFs use the same conformal architecture for $\mathbf{g}$. The basic architecture follows, with input and output channels indicated in brackets. Between every layer, trainable scaling and shift operations were applied.

$$\begin{aligned}
\mathbf{x}\ (3 \times 64 \times 64) \to{}& 8 \times 8\ \text{Householder Conv}\ (3, 192) \\
\to{}& 1 \times 1\ \text{Conditional Orthogonal Conv}\ (192, 192) \\
\to{}& \text{Squeeze}\ (192, 3072) \\
\to{}& \text{Orthogonal Transformation}\ (3072, m) \\
\to{}& \mathbf{u}\ (m)
\end{aligned}$$

**Training.** The sequential baseline for $m = 64$ required a 200-epoch manifold-warmup phase for the reconstruction loss to converge. Otherwise, for the sequential baseline and sequential CEF, $\mathbf{g}$ was trained with a reconstruction loss in a 50-epoch manifold-warmup phase. We then trained $\mathbf{h}$ in all cases to maximize likelihood for 1000-epochs. The joint CEF was trained with the mixed loss function in Eq. (7) for 1000 epochs. All models used weights of 0.01 for the likelihood and 100000 for the reconstruction loss.

Each model was trained on a single Tesla V100 GPU using the Adam optimizer [34] with learning rate $1 \times 10^{-3}$, a batch size of 512, and cosine annealing [47].

## C.3 MNIST

**Models.** All MNIST models use the same architecture for their $\mathbf{h}$ components: a simple 8-layer rational-quadratic neural spline flow with 3 residual blocks per layer and 512 hidden channels each. It is applied to flattened data of dimension 128.

The baseline's embedding $\mathbf{g}$ is a rational-quadratic neural spline flow network of 3 levels, 3 steps per level, and 3 residual blocks per step with 64 hidden channels each. The output is flattened and transformed with an LU-decomposed linear layer, then projected to 128 dimensions.

Both CEFs use the same conformal architecture for $\mathbf{g}$. The basic architecture follows, with input and output channels indicated in brackets. Between every layer, trainable scaling and shift operations

were applied.

$$\mathbf{x} \ (3 \times 64 \times 64) \rightarrow 8 \times 8 \ \text{Householder Conv} \ (1, 64)$$
$$\rightarrow 1 \times 1 \ \text{Conditional Orthogonal Conv} \ (64, 64)$$
$$\rightarrow \text{Squeeze} \ (64, 1024)$$
$$\rightarrow \text{Orthogonal Transformation} \ (1024, 128)$$
$$\rightarrow \mathbf{u} \ (128)$$

**Training.**    For the sequential baseline and sequential CEF, $\mathbf{g}$ was trained with a reconstruction loss in a 50-epoch manifold-warmup phase, and then $\mathbf{h}$ was trained to maximize likelihood for 1000-epochs. The joint CEF was trained with the mixed loss function in Eq. (7) for 1000 epochs. All models used weights of 0.01 for the likelihood and 100000 for the reconstruction loss.

Each model was trained on a single Tesla V100 GPU using the Adam optimizer [34] with learning rate $1 \times 10^{-3}$, a batch size of 512, and cosine annealing [47].

## C.4    CelebA

**Models.**    All CelebA models use the same architecture for their $\mathbf{h}$ components: a 4-level multi-scale rational-quadratic neural spline flow 7 steps per level, and 3 residual blocks per step with 512 hidden channels each. It takes squeezed inputs of $24 \times 8 \times 8$, so we do not squeeze the input before the first level in order to accommodate an extra level.

The baseline's embedding $\mathbf{g}$ is a rational-quadratic neural spline flow network of 3 levels, 3 steps per level, and 3 residual blocks per step with 64 hidden channels each. The output of each scale is reshaped into $8 \times 8$, and the outputs of all scales are concatenated. We then apply an invertible $1 \times 1$ convolution, and project the input down to 1536 dimensions. Since this network is not conformal, joint training is intractable, so it must be trained sequentially.

On the other hand, both CEFs use the same conformal architecture for $\mathbf{g}$. The basic architecture follows, with input and output channels indicated in brackets. Between every layer, trainable scaling and shift operations were applied.

$$\mathbf{x} \ (3 \times 64 \times 64) \rightarrow 4 \times 4 \ \text{Householder Conv} \ (3, 48)$$
$$\rightarrow 1 \times 1 \ \text{Conditional Orthogonal Conv} \ (48, 24)$$
$$\rightarrow 2 \times 2 \ \text{Householder Conv} \ (24, 96)$$
$$\rightarrow 1 \times 1 \ \text{Conditional Orthogonal Conv} \ (96, 96)$$
$$\rightarrow 1 \times 1 \ \text{Householder Conv} \ (96, 96)$$
$$\rightarrow 1 \times 1 \ \text{Orthogonal Conv} \ (96, 24)$$
$$\rightarrow \mathbf{u} \ (24 \times 8 \times 8)$$

**Training.**    For the sequential baseline and sequential CEF, $\mathbf{g}$ was trained with a reconstruction loss in a 30-epoch manifold-warmup phase, and then $\mathbf{h}$ was trained to maximize likelihood for 300-epochs. The joint CEF was trained with the mixed loss function in Eq. (7) for 300 epochs. All models used weights of 0.001 for the likelihood and 10000 for the reconstruction loss.

Each model was trained on a single Tesla V100 GPU using the Adam optimizer [34] with learning rate $1 \times 10^{-4}$, a batch size of 256, and cosine annealing [47].

## D    Reconstructions and Samples

### D.1    Reconstructions

In this section we compare reconstructions from the remaining models omitted in the main text. These were trained on the synthetic ship manifold with 64 dimensions (Fig. 9), and CelebA (Fig. 10).

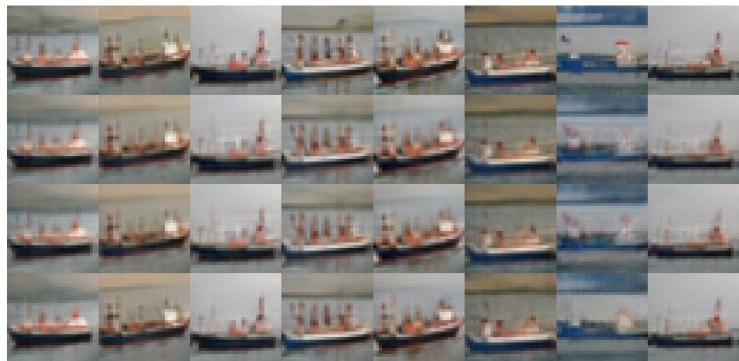

Figure 9: Synthetic 64-dimensional Ship Manifold Reconstructions. From top to bottom: groundtruth samples, joint CEF, sequential CEF, and sequential MF.

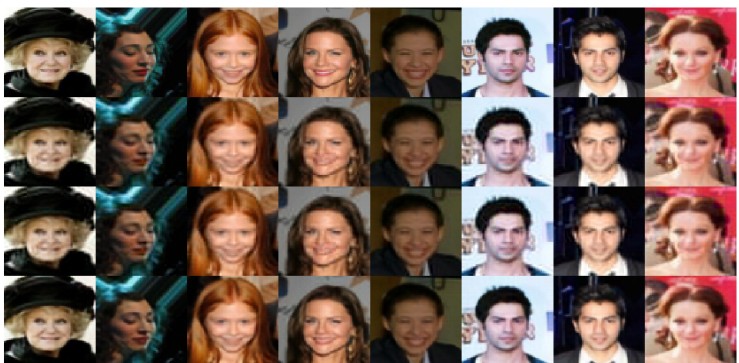

Figure 10: CelebA Reconstructions. From top to bottom: joint CEF, sequential CEF, and sequential MF.

## D.2 Samples

In this section we provide additional samples from all the image-based models we trained. Figs. 11-16 show the synthetic ship manifolds, Figs. 17-19 show MNIST, and lastly Figs. 20-22 show CelebA.

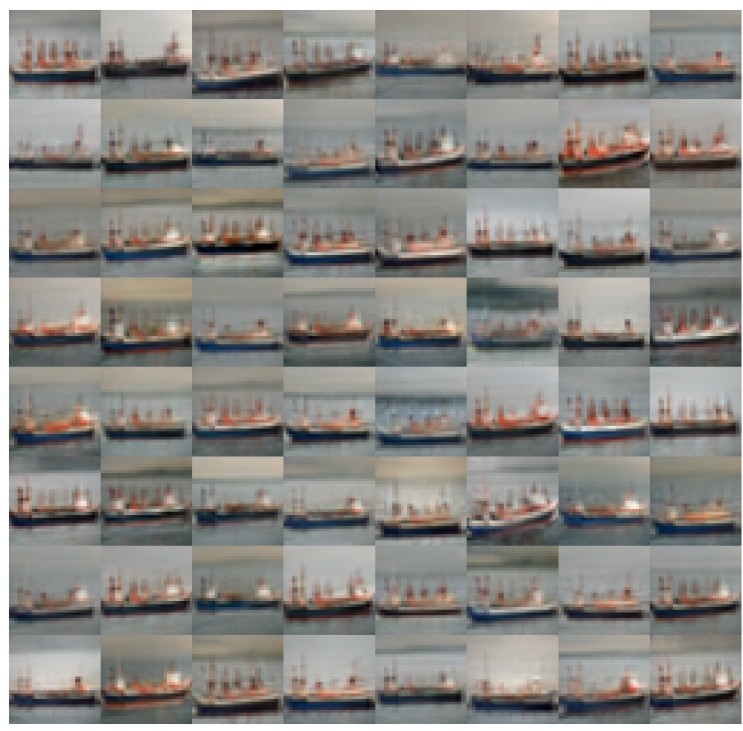

Figure 11: Uncurated Synthetic 64-dimensional Ship Manifold Samples: Joint CEF

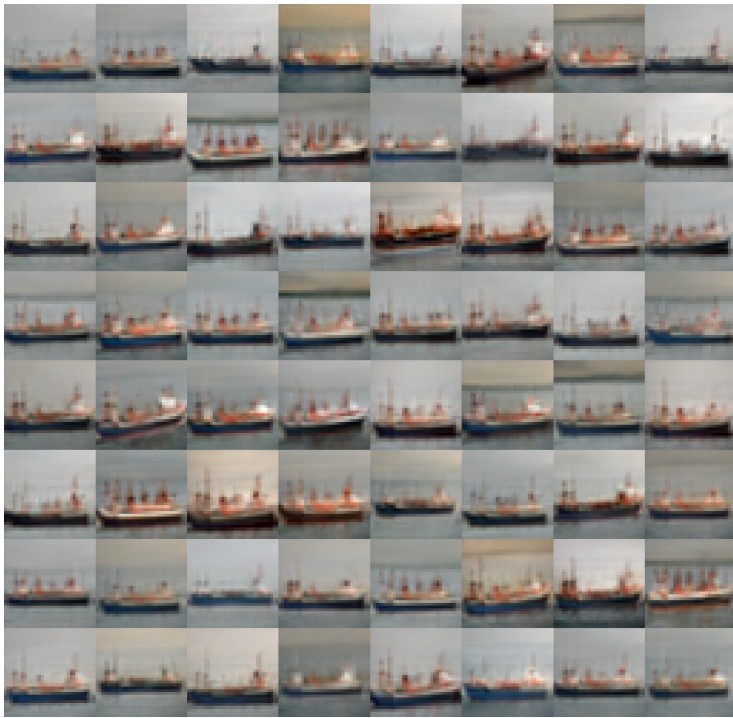

Figure 12: Uncurated Synthetic 64-dimensional Ship Manifold Samples: Sequential CEF

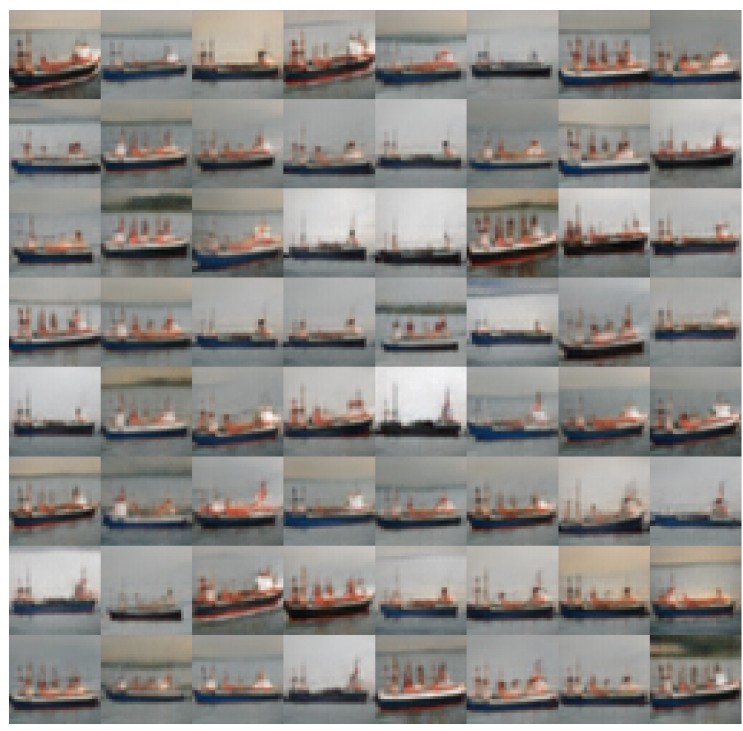

Figure 13: Uncurated Synthetic 64-dimensional Ship Manifold Samples: Sequential MF

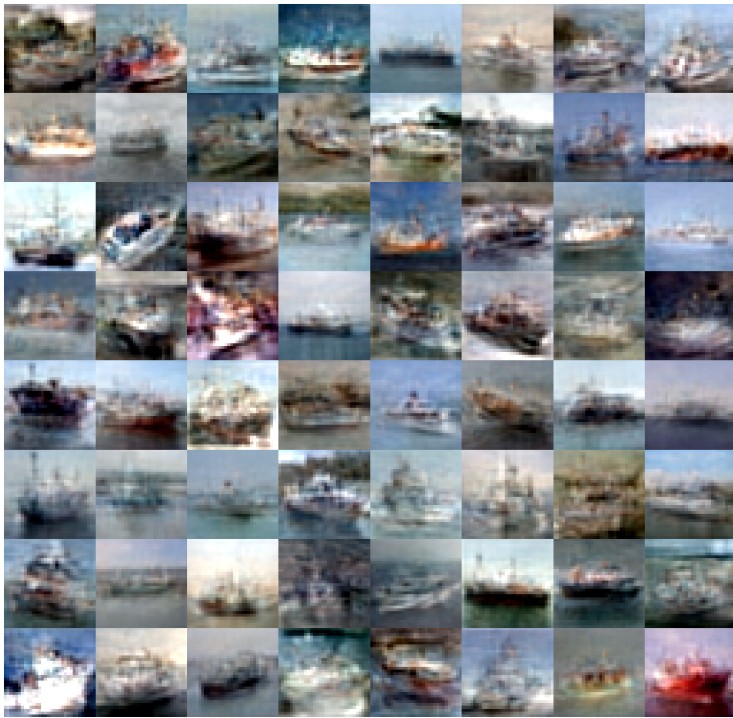

Figure 14: Uncurated Synthetic 512-dimensional Ship Manifold Samples: Joint CEF

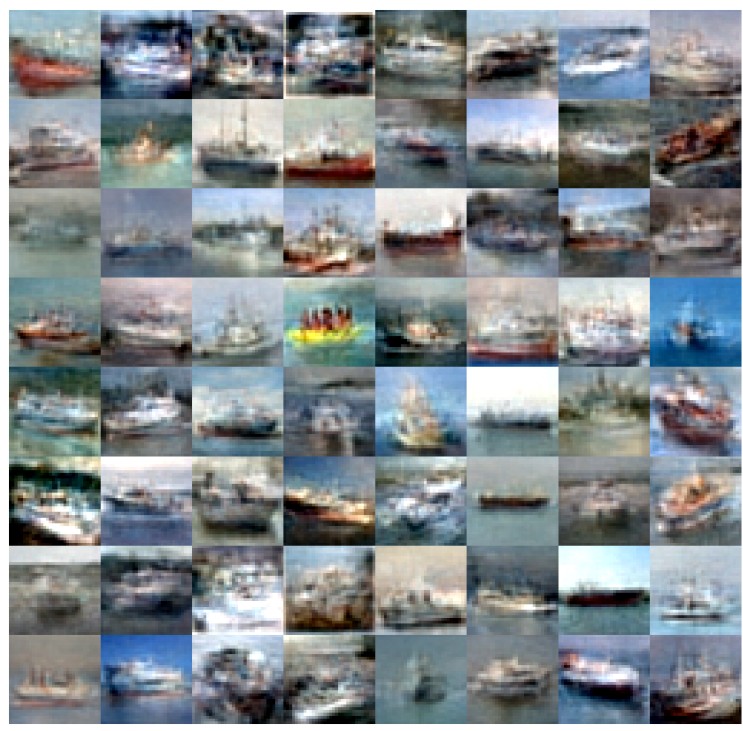

Figure 15: Uncurated Synthetic 512-dimensional Ship Manifold Samples: Sequential CEF

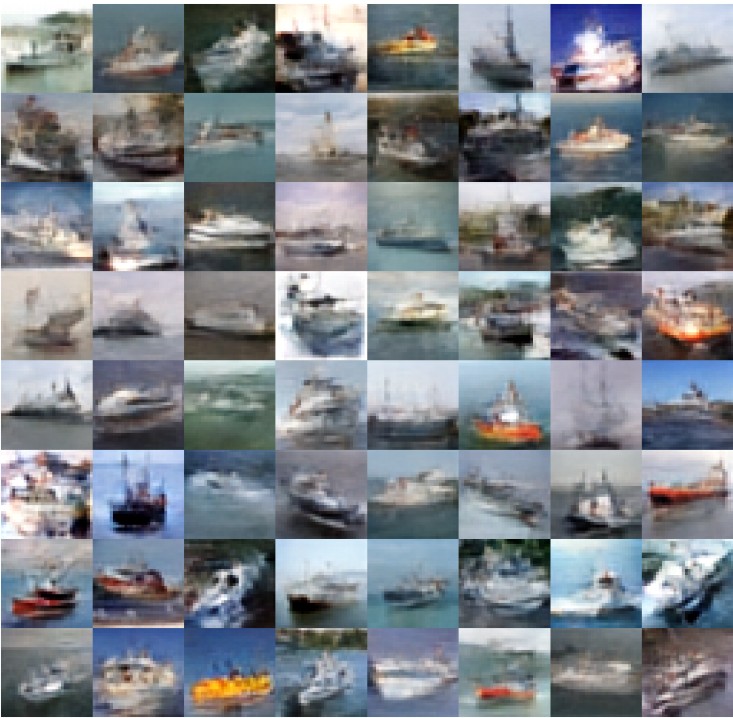

Figure 16: Uncurated Synthetic 512-dimensional Ship Manifold Samples: Sequential MF

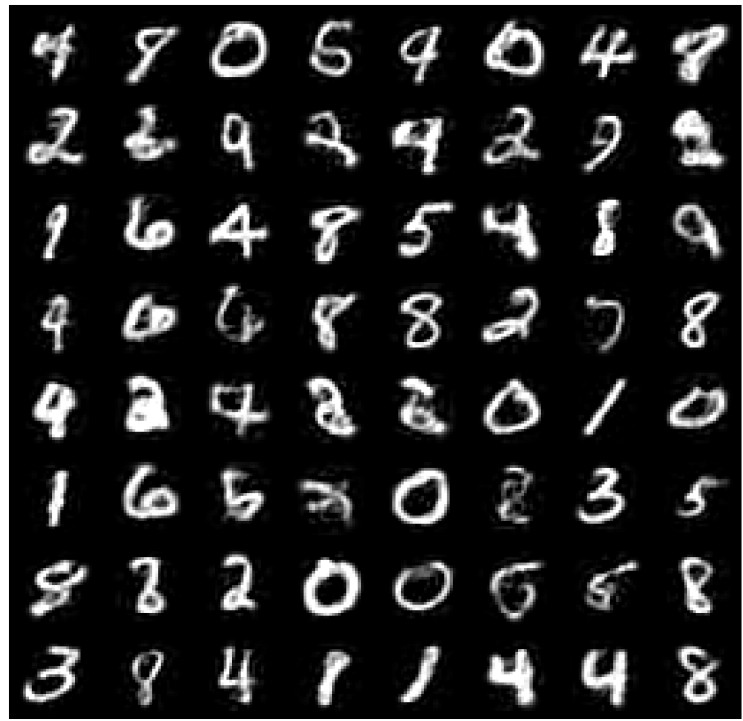

Figure 17: Uncurated MNIST Samples: Joint CEF

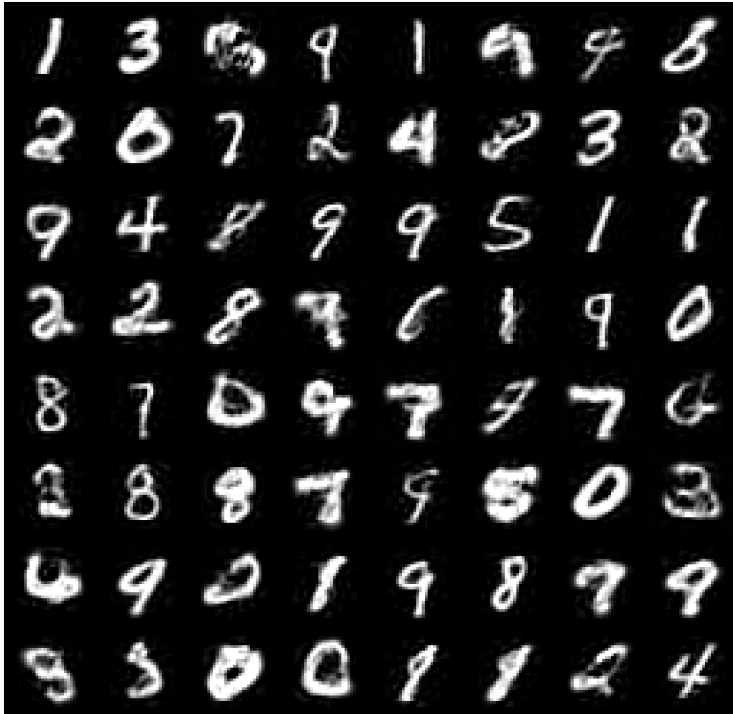

Figure 18: Uncurated MNIST Samples: Sequential CEF

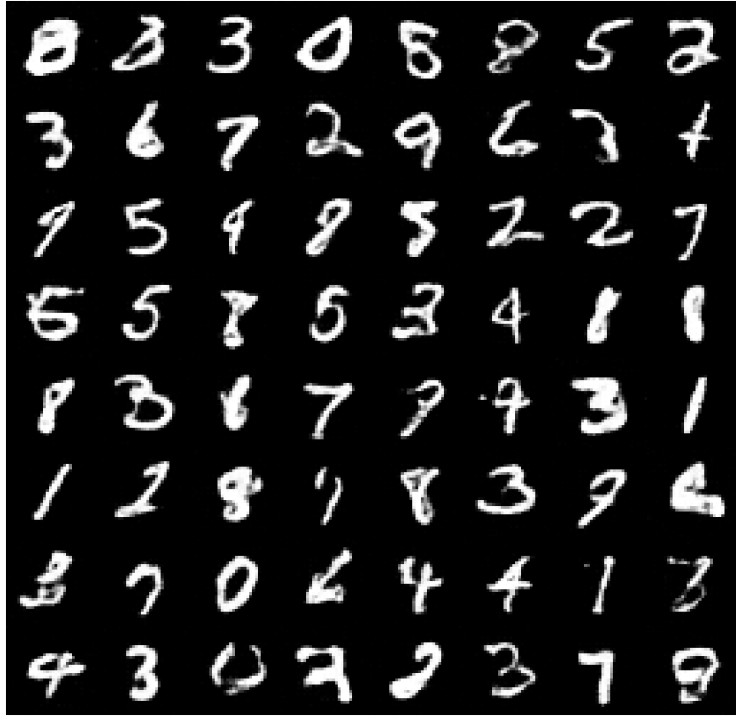

Figure 19: Uncurated MNIST Samples: Sequential MF

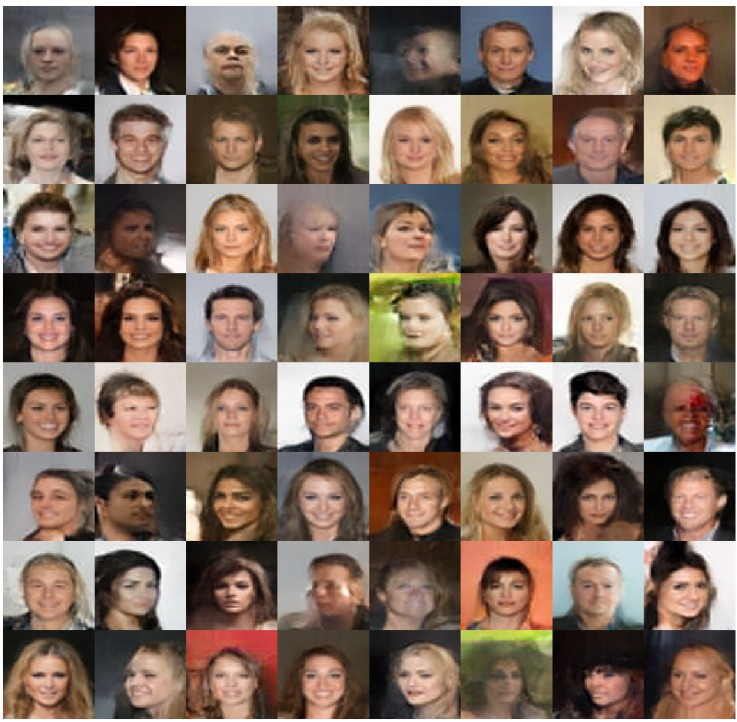

Figure 20: Uncurated CelebA Samples: Joint CEF

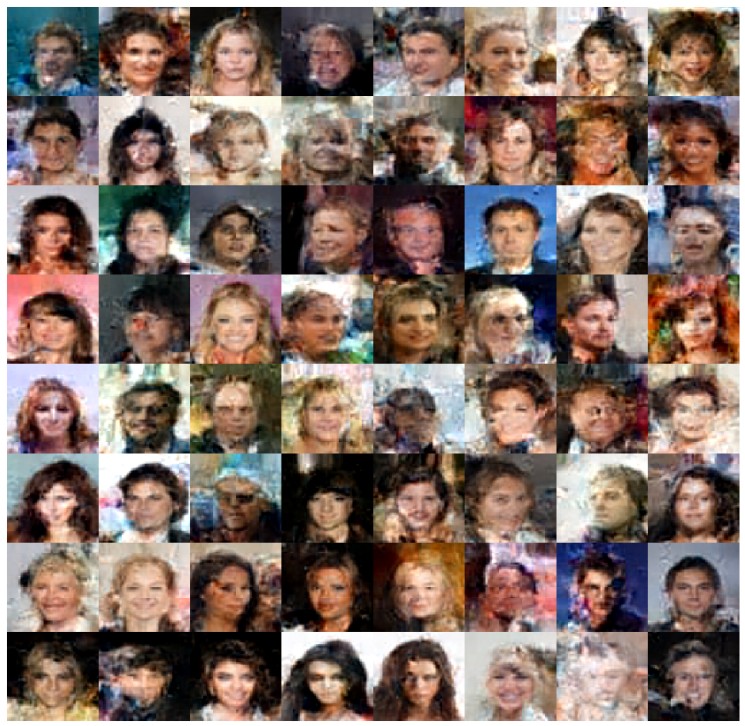

Figure 21: Uncurated CelebA Samples: Sequential CEF

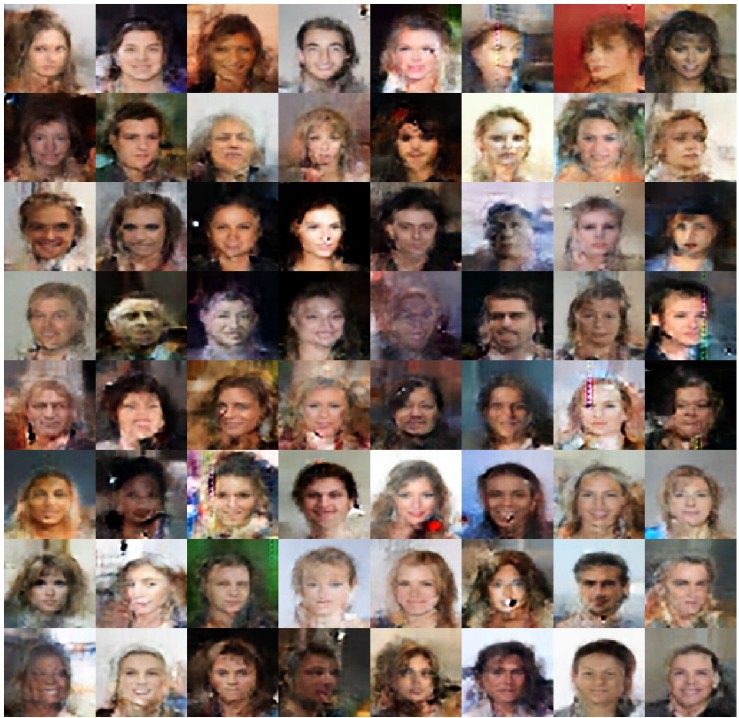

Figure 22: Uncurated CelebA Samples: Sequential MF