# OpenReview forum: "Tractable Density Estimation on Learned Manifolds with Conformal Embedding Flows"
_NeurIPS.cc/2021/Conference — NeurIPS 2021 Poster_

### Official Review · Reviewer_wnDz · 2021-07-07

**Rating:** 5
**Confidence:** 4

**Summary:**

The paper proposes classes of conformal building blocks that can be used to parameterize a homomorphic flow network that can map from low dimensional space to a continuous manifold in high dimension space.

**Limitations And Societal Impact:**

Yes, the author mentioned the major limitation is expressive power.

**Main Review:**

Originality: The paper restricts function g to be a conformal embedding. The k*k convolution is new, the SCT is new to the machine learning field.

Clarity: The paper is clearly written. But I think the novel building blocks like k*k convolution should be detailed discussed.

Quality: The submission is technically sound.

Significance and Concerns:
1. My first concern is more general. The traditional flow models aim to have a tractable likelihood for maximum likelihood estimation and achieves success in a lot of applications. The paper interests in modeling distributions that lie on a low-dimensional manifold. As the author mentioned, the density is no longer well defined (the distribution is not absolutely continuous with respect to Leabage measure), so the maximum likelihood estimation (MLE)  and the KL divergence are not well defined,  so why we need to construct a function with tractable log density in the first place when the density is not defined? Furthermore, the method has to use L2 reconstruction loss as a regularization to alleviate the ill-defined MLE (equation 7), does this objective provide a consistent estimation, and what distance it is minimizing? Why not just directly use a distance which is well defined for the singular distribution like e.g. Wasserstein distance. Although I understand there are several papers that use this setting but the benefit of such a setting is unclear to me which makes me less confident about the motivation of the paper.

2. The paper restricted the function g to be conformal, and propose several building blocks. However, the experiments are limited and cannot support the significance of the proposed methods. The samples of the CelebA experiments are far from good, which maybe because of the limited function power or the training objective with the l2 norm. The reconstruction quality is less meaningful for generative models, it can only verify the invertibility.  I think more experiments and comparisons should be conducted to support the usefulness of the framework. I can understand the  CelebA dataset may not satisfies the model assumption (lies exactly on a low-dimensional manifold ) and the intrinsic dimension is unknown. So I suggest the training data can be generated from a trained GAN model, so the intrinsic dimension is (approximately) equal to the dimension of the latent space and the generated samples will lie on a low dimensional manifold.



**Time Spent Reviewing:**

6 hours

---

> ### Author Response · Authors · 2021-08-10
> **Response to Reviewer wnDz**
>
> Thank you for your valuable feedback and questions. We're glad you found our work original and well-written.
>
> You pointed out that the novel building blocks like the $k\times k$ convolution and special conformal transformation should be discussed in more detail. We will update the methods section with more specific implementation details.
>
> 1. Your first set of concerns was about the general motivation for the research setting of injective flows. To motivate it in a rigorous way, we will include an appendix from the measure theoretic perspective based on the following analysis:
>
>     > "As the author mentioned, the density is no longer well defined (the distribution is not absolutely continuous with respect to Leabage measure), so the maximum likelihood estimation (MLE) and the KL divergence are not well defined, so why we need to construct a function with tractable log density in the first place when the density is not defined?"
>
>      Thank you for bringing this up; it is a useful point of discussion. We'd like to point out that when the data distribution $P$ is on an embedded submanifold, the density *is* actually well-defined, just with respect to a different base measure: the Riemannian measure of the submanifold. This is the natural choice of base measure used in the normalizing flow literature, eg. [14, 33, 37].
>
>     This change in base measure is exactly why we need an injective flow. If the model were a standard flow $Q_\theta$, $P$ would not be absolutely continuous with respect to $Q_\theta$. Pathologically, KL-minimization and MLE are distinct here, but neither bodes well for the modeller: KL is undefined, and MLE produces the problematic results discussed in lines 74-81.
>
>     On the other hand, when $Q_\theta$ is an *injective* flow, the KL can be well-defined. We just need $Q_\theta$ to be absolutely continuous with respect to the submanifold's Riemannian measure; in other words, the model and data manifolds must line up. This is exactly what the reconstruction loss is doing - finding the correct base measure.
>
>     In practice, the model and data manifolds might not *exactly* line up, but such is the coarse nature of many machine learning settings. The syncing of base measures to make the KL theoretically well-defined occurs, in general, only in the non-parametric limit of the manifold learner $g$, but similar requirements also crop up often in deep learning, and are ubiquitous in the GAN literature. Fortunately, as with an imperfect GAN discriminator, a slightly incorrect manifold still yields a useful density training signal because the loss is (almost-everywhere) continuous with respect to $g$'s parameters.
>
>     Once the base measures of $p$ and $q_\theta$ are in sync, one can show $P$ is absolutely continuous with respect to $Q_\theta$, so we can train using MLE/KL (which are once again equivalent). *This* is where a tractable log density, our work's contribution, becomes useful. Once the model is trained, the tractable log density can also be used for downstream tasks, such as typicality testing and outlier detection.
>
>     > "Furthermore, the method has to use L2 reconstruction loss as a regularization to alleviate the ill-defined MLE (equation 7), does this objective provide a consistent estimation, and what distance it is minimizing?"
>
>     Good question - the objective is probably not a known distance metric because it depends not only on $Q_\theta$ and $P$, but also on the structure of the injective flow. On the bright side, since the expected reconstruction loss is non-negative and non-degenerate, the manifold will be learned perfectly if it is minimized with enough capacity.
>
>     *Sequential* training, where we first train the manifold then the density, should thus provide consistent estimation if the network has enough capacity. When the manifold has been trained perfectly, the model density is simply being learned via MLE. We expect then that in every situation there exists a sufficiently high $\alpha$ for which *joint* training (eq. 7) is consistent, though we leave investigation of this to future work.
>
>     The repeated requirement of "enough capacity" raises the question of how flexible conformal embeddings can be. We suspect that a piecewise conformal embedding can achieve some kind of "universal manifold approximation" because any piecewise linear manifold can be expressed as the union of conformally embedded pieces. We have not nearly achieved the full capacity of conformal embeddings, and this work is partially meant as an entreaty to other researchers to find more expressive conformal embeddings.
>
>     > "Why not just directly use a distance which is well defined for the singular distribution like e.g. Wasserstein distance."
>
>     We agree that, if tractable, the Wasserstein distance would be a more natural objective. However, in practice it requires adversarial training to be estimated [A], and can't really be estimated without bias as its sample complexity is exponential in dimension [B]. On the other hand, CEFs provide MLE, which is known to provide stable training and has a straightforward interpretation. We see WGANs and injective flows as different approaches to the same problem.
>
> 2. You shared the other reviewers' concerns about the experiment section. We will expand this section in the final paper - please see the general comment and the [preliminary results](https://sites.google.com/view/cefs-neurips2021/home "CEFs NeurIPS 2021 Responses") for details. Thank you for recommending experiments on a GAN manifold - we are running some with a CIFAR-10-trained GAN.
>   We also point out that though the reconstruction quality may be less important for typical flow models, it is of special interest for injective flow models because it shows how well the manifold has been learned. This is the main hump for CEFs to overcome as a paradigm because they represent a structural restriction on the manifold learner $g$.
>
>
> [A] Arjovsky, M., Chintala, S. & Bottou, L.. (2017). Wasserstein Generative Adversarial Networks. Proceedings of the 34th International Conference on Machine Learning, in Proceedings of Machine Learning Research 70:214-223 Available from http://proceedings.mlr.press/v70/arjovsky17a.html .
>
> [B] Arora, S., Ge, R., Liang, Y., Ma, T. & Zhang, Y.. (2017). Generalization and Equilibrium in Generative Adversarial Nets (GANs). Proceedings of the 34th International Conference on Machine Learning, in Proceedings of Machine Learning Research 70:224-232 Available from http://proceedings.mlr.press/v70/arora17a.html .

---

> > ### Comment · Reviewer_wnDz · 2021-08-11
> > **Concerns about the L2 regularizer**
> >
> > 1. We usually define the density is respect to Lebesgue measure, I agree that the author can define a relative density with respect to a manifold measure. In this case, as the author also mentioned, the data manifold and the model manifold need to be perfectly aligned, which has probably zero (when the intrinsic dimension of the manifold is smaller than the ambient dimension) (lemma 2, [1]). Also, perfect alignment needs knowing the true intrinsic dimension, which is problematic.
> > 2. Therefore, the author argues adding the L2 regularizer aims to make two manifolds align,  which is not very obvious to me.  The regularizer force the point-wise alignment but not manifold-wise. It makes the training more like a  (regularized) auto-encoder. And the experiment also somehow supports this: the reconstruction is good but the sample is still poor. I think if there is a manifold matching regularizer, and under that regularizer, the proposed method can work (and the regularizer goes to 0), then the whole story is more convincing.
> > 3. I am still not convinced that the reconstruction is too meaningful, an auto-encoder with L2 regularizer can also have perfect reconstruction. Additionally, the training objective does look like an auto-encoder.
> > 4.  I personally don't think it is a coarse nature of many machine learning settings.  In the modern likelihood-based generative model, like big VAE, the decoder is always a discrete distribution (or the model is not an implicit model), then everything is well defined. Or in GAN, the problem is alleviated by using some other distance like Wasserstein or some MMD variants.
> > 5. Although I agree with the author WGAN also has its problem, but it at least has better samples. However, one may argue that WGAN is more good at estimating the support but not the 'density' on the manifold [2], which the proposed method may have a winning, but experiments need to be done to show this.
> >
> > I will raise my rating to 5 due to the respectful work done during the rebuttal.
> >
> > I think I will give an acceptance when  any one of the following 3 points is met or solved:
> > 1. Theory: the unsatisfying l2 regularizer, see the above comment.
> > 2. Practice: good sample quality, at least as good as the training sample generated from GAN. In this case, you know the intrinsic dimension so the problem should be easier.
> > 3.  Relative density: show the proposed method can have better relative density estimation comparing to Wasserstein GAN. Maybe some low-dimensional constructed problems where the true relative density is known.
> >
> >
> >
> >
> >
> > [1] Arjovsky, Martin, and Léon Bottou. "Towards principled methods for training generative adversarial networks." arXiv preprint arXiv:1701.04862 (2017).
> >
> > [2] Arbel, Michael, Liang Zhou, and Arthur Gretton. "Generalized energy based models." arXiv preprint arXiv:2003.05033 (2020).

---

> > > ### Author Response · Authors · 2021-08-14
> > > **Response to Concerns About L2 Regularizer**
> > >
> > >  Once again, thank you for raising these lines of inquiry. We feel this discussion is truly strengthening the foundation of the work. We also appreciate the increase in score along with the prompt and thoughtful response.
> > >
> > > ### L2 Regularizer
> > > Here is why the L2 regularizer encourages manifold alignment. We will include this in an appendix.
> > >
> > > Let $f_\theta: \mathbb{R}^m \to \mathbb{R}^n$ be an injective flow ($m < n$) with a smooth left-inverse $f_\theta^\dagger$ as defined on lines 82-88. By construction, the image $\mathcal{M}_\theta = f_\theta(\mathbb{R}^m)$ of the flow is a Riemannian submanifold of $\mathbb{R}^n$; we call $\mathcal{M}_\theta$ the *model manifold*.
> > >      Let $P$ be a data distribution supported by an $m$ - dimensional Riemannian submanifold $\mathcal{M}_d$; we call this the *data manifold*. Suppose furthermore that $P$ admits a probability density with respect to the Riemannian measure of $\mathcal{M}_d$.
> > >
> > > - **Proposition.** We claim that $R(\theta) := E_{x \sim P}||x - f_\theta(f_\theta^\dagger(x))||_2 = 0$ if and only if $\mathcal{M}_d \subseteq \text{cl}\left(\mathcal{M}_\theta\right)$. Since the reconstruction loss $R(\theta)$ is continuous, we infer from this that $R(\theta) \to 0$ will bring $\mathcal{M}_\theta$ and $\mathcal{M}_d$ into alignment (except possibly for the $P$-null set $\mathcal{M}_d \cap \partial\mathcal{M}_\theta$).
> > >
> > > - **Proof.**
> > >    $(\implies)$ For the forward direction, suppose $R(\theta) = 0$. For a single point $x \in \mathbb{R}^n$, it's true by the definition of $f^\dagger$ that $x = f_\theta( f_\theta^\dagger(x))$ if and only if $x \in f(\mathbb{R}^m) = \mathcal{M}_\theta$. Put differently,
> > >
> > >    $\mathcal{M}_\theta = \\{ x \in \mathbb{R}^n: x = f_\theta( f_\theta^\dagger(x) ) \\}$,
> > >
> > >    so a reconstruction error of zero implies $P(\mathcal{M}_\theta) = 1$. This means that $P$'s support must by definition be a subset of $\text{cl}\left(\mathcal{M}_\theta\right)$. It follows that $\mathcal{M}_d \subseteq \text{cl}\left(\mathcal{M}_\theta\right)$.
> > >
> > >    $(\impliedby)$ For the reverse direction, note that if $\mathcal{M}_d \subseteq \text{cl}\left(\mathcal{M}_\theta\right)$, then $\mathcal{M}_d \setminus f_\theta(\mathbb{R}^m)$ has measure zero in $\mathcal{M}_d$, so $f(f^\dagger(x)) = 0$ $P$ -almost surely. This fact yields $R(\theta) = 0$.
> > >
> > >  This ends the proof.
> > >
> > > This is why $R(\theta)$ should be considered a manifold matching regularizer. This is the reason we emphasize reconstruction loss - we treat it as an indicator of manifold alignment. [The arXiv version of [3]](https://arxiv.org/pdf/2003.13913.pdf) also goes into detail to motivate the reconstruction loss - in particular see Figure 3 and section 3A. Hence Lemma 2 in [A] is less of a concern for injective flows. Lemma 2 refers to arbitrary manifolds perturbed by noise to emulate manifolds in nature, but in our setting, manifold alignment has been explicitly encouraged by the reconstruction loss.
> > >
> > > We agree that not knowing the intrinsic dimension is a problem, and determining it is a topic of research in its own right - eg. [B]. Such methods could be used as a precursor step before designing and training a CEF. Moreover, if the dimensionality of the model is higher than the data, the model's manifold can contain the data manifold, and hence maximum likelihood estimation can still be performed as with a standard normalizing flow. We would still in this case encounter the theoretical issues discussed in lines 74-81. Standard flows are a special case of this in which the model manifold is the whole ambient space.
> > >
> > > ### Image Quality
> > >
> > > We expect that the model can achieve sample quality close to the quality of the reconstructions, but we do not have experiments to support this. If it's possible, it will require a deeper and possibly more complex backbone flow $h$ with a longer training time. If we are able to improve on sample quality over time, we will include it in a later version of the paper.
> > >
> > > ### Comparison with WGANs
> > >
> > > We appreciate your suggestion to compare our method to WGANs. We tested our method against Wasserstein training on 12800 points sampled from a 2D Gaussian mixture embedded as a plane in 3D space. Both experiments shared one architecture - $g \circ h$, where $h$ was a series of 3 Glow steps in 2D ([24] Fig. 2a), and $g$ padded the output of $h$ into 3D, and applied an orthogonal transformation.
> > >
> > > We found that likelihood-based training with an L2 reconstruction regularizer learned the manifold visually perfectly and the density well, but training adversarially with the Wasserstein objective was not able to perfectly match the manifold and the density was poor. To enforce the Lipschitz constraint we tried both weight clamping [C] and gradient penalties [D] with numerous hyperparameter combinations, but we were not able to achieve reasonable-looking densities with the WGAN. The visual quality and log-likelihood tended to improve initially then erode as training progressed. This behaviour corresponds to the observations of [E], in which a flow with full support is trained adversarially.
> > >
> > > This experiment suggests that, as you hypothesized, injective flows provide better density estimation than GANs on manifolds. Images of the learned manifolds and densities can be found [here](https://sites.google.com/view/cefs-neurips2021/home "CEFs NeurIPS 2021 Responses"). We will include these results in the final copy.
> > >
> > >
> > > [A] Martin Arjovsky and Léon Bottou. (2017). Towards principled methods for training generative adversarial networks. arXiv preprint arXiv:1701.04862.
> > >
> > > [B] Elizaveta Levina and Peter J. Bickel. (2004). Maximum Likelihood Estimation of Intrinsic Dimension. In Advances in Neural Information Processing Systems, volume 17.
> > >
> > > [C] Martin Arjovsky, Soumith Chintala, and L´eon Bottou. (2017). Wasserstein generative adversarial networks. In Proceedings of the 34th International Conference on Machine Learning, pages 214–223.
> > >
> > > [D] Ishaan Gulrajani, Faruk Ahmed, Martin Arjovsky, Vincent Dumoulin, and Aaron Courville. Improved training of wasserstein GANs. arXiv preprint arXiv:1704.00028, 2017.
> > >
> > > [E] Aditya Grover, Manik Dhar, and Stefano Ermon. (2018). Flow-GAN: Combining maximum likelihood and adversarial learning in generative models. In Proceedings of the 32nd AAAI
> > > Conference on Artificial Intelligence.

---

> > > > ### Comment · Reviewer_wnDz · 2021-08-14
> > > > **Concerns remain**
> > > >
> > > > 1. Sorry, maybe my previous comment about L2 regularizer isn't clear enough.
> > > > I do agree that perfect L2 will make manifold matching, but it additionally encourages point-wise matching.  Since if I just have L2 regularizer as my training criterion,  I can also have perfect reconstruction when the L2 goes 0, which makes the reconstruction experiment not very meaningful.
> > > > Now that we want to build a model and be able to sample from the model, we have to make the embeddings to be close to some prior distribution, like the first term in the proposed loss.  Therefore, the sample quality is the only way to judge the model is good or not. But the model cannot generate good CelebA samples, which is much simpler than CIFAR or other nature data.
> > > >
> > > >
> > > > 2. "the model's manifold can contain the data manifold, and hence maximum likelihood estimation can still be performed as with a standard normalizing flow" this is not true. Since the invertibility of the flow function, the model will not be able to degenerate (not like GAN), so the intrinsic dimension of the flow model will be equal to the dimension of the Gaussian prior. In this case, the intrinsic dimension of the model will be larger than the intrinsic dimension of the data, the model will never learn the data distribution correctly.
> > > >
> > > > 3. The density plot does look better than WGAN, which is encouraging.  However,  we can still see some connected paths between disconnect manifolds, maybe due to the topology-preserving of the injective flow?
> > > >
> > > > It seems the proposed model can work well in low-dimensional data but not in high-dimensional data, I will discuss with AC to see if it is a problem.

---

> > > > > ### Author Response · Authors · 2021-08-16
> > > > > **Response to Concerns**
> > > > >
> > > > > Thank you for the prompt response and continued engagement! We greatly appreciate your ongoing effort.
> > > > >
> > > > > 1. Would you be able to clarify your remaining concerns about the L2 reconstruction term?
> > > > >
> > > > >    -  It appeared your original concern was whether the L2 reconstruction term encourages manifold alignment:
> > > > >
> > > > >       > "Therefore, the author argues adding the L2 regularizer aims to make two manifolds align, which is not very obvious to me. The regularizer force the point-wise alignment but not manifold-wise ... I think if there is a manifold matching regularizer, and under that regularizer, the proposed method can work (and the regularizer goes to 0), then the whole story is more convincing."
> > > > >
> > > > >       We demonstrated that L2 reconstruction is indeed manifold matching theoretically above and empirically in the paper's sphere example and the [WGAN comparison](https://sites.google.com/view/cefs-neurips2021/home "CEFs NeurIPS 2021 Responses"). The latter experiment also provided a reason to use our objective over Wasserstein distance. Is there anything else we can discuss to this end?
> > > > >
> > > > >    - You also emphasize that L2 reconstruction encourages point-wise matching in addition to manifold matching. If this is a concern of yours, can you expand on why?
> > > > >
> > > > >    - You expressed that the model's reconstructions are not meaningful because a model can have good reconstructions without actually having good sample quality.
> > > > >
> > > > >      While we agree reconstructions do not tell the full story, they are a visual indicator of manifold alignment as shown by the proof above, which is why we take them to be informative.
> > > > >
> > > > >      To recap, our model is simultaneously performing two tasks: (a) maximum likelihood estimation within the model manifold and (b) aligning the model and data manifolds. These correspond to the two loss terms in eq. (7):
> > > > >
> > > > >      $\mathcal{L} = \mathbb{E}_{\mathbf{x} \sim p^*_\mathbf{x}}[\underbrace{-\log p_\mathbf{x}(\mathbf{x})}_\text{(a)} + \alpha \underbrace{\left|\left| \mathbf{x} - \mathbf{g}\left(\mathbf{g}\left(\mathbf{x}\right)\right)\right|\right|}_\text{(b)}]$
> > > > >
> > > > >      The model's capacity for (a) can be improved by increasing the capacity of the backbone flow $h$. We know there are flow models capable of learning complex densities, so (b) is the more pertinent question for our model. (b) can be evaluated by observing reconstructions. Because of this, we explicitly used a small backbone $h$ (which we ambiguously referred to as a stump: we will be more specific in the final version) due to computational constraints.
> > > > >
> > > > >      Since the fidelity of the learned distribution comes from the model's ability to satisfy (a) and (b) together, and the L2 reconstruction term does encourage (b) as shown, it is not clear to us why it should be singled out as the cause of perceived issues with sample quality.
> > > > >
> > > > > > 2. "the model's manifold can contain the data manifold, and hence maximum likelihood estimation can still be performed as with a standard normalizing flow" this is not true ... In this case, the intrinsic dimension of the model will be larger than the intrinsic dimension of the data, the model will never learn the data distribution correctly.
> > > > >
> > > > > We agree the model will learn the distribution incorrectly and apologize for the ambiguity. We would like to clarify that this is the topic of the lines 74-81 we referenced in the response. The purpose of our comment was just to point out that $\mathbb{E}_{x \sim P} \left[\log q_\theta(x)\right]$ can still be optimized, and that this is the same as training a standard normalizing flow on data with low intrinsic dimension.
> > > > >
> > > > > This can produce decent samples, as we see with many flow models trained on image data, but it can't learn the correct distribution. We will make this pathological situation more clear in the paper.
> > > > >
> > > > > > 3. The density plot does look better than WGAN, which is encouraging. However, we can still see some connected paths between disconnect manifolds, maybe due to the topology-preserving of the injective flow?
> > > > >
> > > > > We are glad you find this encouraging, and hope it meets the path to acceptance you suggested: showing improved relative density estimation over Wasserstein training.
> > > > >
> > > > > We note that the support of WGANs, along with any other generative model whose distribution is the continuous pushforward of a normal distribution, must also be connected. This is not just an issue for flow models.

---

> > > > > > ### Comment · Reviewer_wnDz · 2021-08-17
> > > > > > **Why the L2 concerns me.**
> > > > > >
> > > > > > I give a formal explanation of my previous concerns.
> > > > > >
> > > > > >  Assumes encoder function f and decoder function g, (g can be constrained to be the pesudo inverse of f, which is used in the paper). The data distribution is $P_d$.  Let's consider 2 models:
> > > > > >
> > > > > > ### Naive Autoencoder
> > > > > > the loss is $\int |x-g(f(x))|_2 dP_d$. This will have a perfect reconstruction result, but no generation.
> > > > > >
> > > > > > ### Regularized Autoencoder
> > > > > > the loss is $\alpha \int |x-g(f(x))|_2 dP_d+ D(q(z)|| p(z))$.  Where $q(z)=\int \delta(z-f(x))dP_d$. This model is also widely used in many models, such as [1],[2].  The model is $p_\theta(x)=\int \delta(x-g(z))p(z)dz$. This will also have a perfect reconstruction result if alpha is large. The generation will also be good if the emperical $q(z)$  maches $p(z)$ well, which depends on the divergence choice.
> > > > > >
> > > > > > The proposed method falls in this category, with the $D$ to be the KL divergence. $D(q(z)||p(z))=-H(q(z))- \int q(z) \log p(z)dz=-H(p_d)+J- \int p(z=f(x))dP_d(x)$. The first entropy term $H(p_d)$ is constant, the  second J term is the log-determinant term in the paper.  Thus, the training objective is exactly same as the proposed method.  Comparing to [1][2], the difference is that the proposed method explicitly constructs the pseudo inverse $f=g^{\dagger}$.
> > > > > >
> > > > > > Then I realize papers [1] and [2]  all have good reconstructions and generations, and also can generate good samples when data lie on a low-dimensional manifold (like the additional experiment done in the rebuttal, sorry I didn't realize this point in the previous rebuttal round).   However, compared to [1],[2], the generated images are much worse, this makes me further concerned about the usefulness of the paper.  Therefore, I cannot give an acceptance at this stage. I will discuss with AC about my concerns.
> > > > > >
> > > > > > Overall, I do agree that modeling distributions on manifolds using flows is interesting, but the use of the L2 regularizer makes me unsatisfied, it makes the method more like a regularized auto-encoder with worse performance.  Therefore in the previous round, I am trying to think (and ask) about if there is any regularizer that only matches the support but not point-wise to replace the  L2 loss, which will make the proposed method different from [1]][2], and has its contributions in a different competition track.
> > > > > >
> > > > > > ## Minors:
> > > > > > "We note that the support of WGANs, along with any other generative model whose distribution is the continuous pushforward of a normal distribution, must also be connected. This is not just an issue for flow models." This is not true as well, push forward operation will not necessarily preserve the topology, it is more like a problem specific to the bijective flow (flow function is a homeomorphism) [3].
> > > > > >
> > > > > > Thanks
> > > > > >
> > > > > > ## Reference
> > > > > > [1] Makhzani, Alireza, et al. "Adversarial autoencoders." arXiv preprint arXiv:1511.05644 (2015).
> > > > > > [2] Tolstikhin, Ilya, et al. "Wasserstein auto-encoders." arXiv preprint arXiv:1711.01558 (2017).
> > > > > > [3] Cornish, Rob, et al. "Relaxing bijectivity constraints with continuously indexed normalising flows." International Conference on Machine Learning. PMLR, 2020.

---

> > > > > > > ### Author Response · Authors · 2021-08-20
> > > > > > > **CEF to Regularized Autoencoder Comparison and Continuous Pushforwards**
> > > > > > >
> > > > > > > ## L2 Reconstruction Term: Comparison between CEFs and AAEs/WAEs
> > > > > > >
> > > > > > > Thank you for the helpful clarification. While we agree with your observation that the objectives of AAEs[A]/WAEs[B] can be modified to look like the joint CEF objective, there are substantial differences between regularized autoencoders and CEFs. In spite of any superficial similarity, the training objectives have different interpretations and must be handled in different ways due to their distinct architectures. As a point of comparison, the architectural constraints of normalizing flow models make them a distinct line of generative modelling research, even when they are trained with an L2 reconstruction term.
> > > > > > >
> > > > > > > ### Motivation and Model Construction
> > > > > > >
> > > > > > > CEFs were designed with the goal of tractable density estimation. Tractable densities are interesting (in our opinion) because they provide maximum-likelihood estimation and can also be used for downstream tasks. Other models with tractable densities, such as normalizing flows, implicitly assume the data has full support. CEFs relax this assumption to low-dimensional submanifolds, which is more realistic in some natural settings.
> > > > > > >
> > > > > > >   There are two main purposes to our work:
> > > > > > >   1. To disseminate the core idea that conformal embeddings can be used to build manifold-supported flow models with tractable densities.
> > > > > > >   2. To provide some conformal building blocks out of which such models can be built.
> > > > > > >
> > > > > > > In contrast, neither AAEs nor WAEs provide likelihood estimates due to their architectures. You are right to point out that $g = f^\dagger$ is a major difference. The other structural difference is that CEFs have a tractable $\log\det J^T J$ term. Both are necessary for density estimation.
> > > > > > >
> > > > > > > ### Training
> > > > > > >
> > > > > > > While training a CEF, we add an L2 reconstruction loss to the density. As discussed in our original submission, we followed [3] (NeurIPS 2020) in doing so. They justify this loss extensively in their [arXiv version](https://arxiv.org/pdf/2003.13913.pdf) (section 3A), and we have formalized this justification in the response above.
> > > > > > >
> > > > > > > We agree that our likelihood maximization term can be rewritten as KL divergence minimization in latent space, making the loss look like a regularized autoencoder's. However, as left-invertibility is removed when the model is changed to an autoencoder, this equivalence is lost; the autoencoder's latent KL-minimization term would no longer be equivalent to MLE on the manifold.
> > > > > > >
> > > > > > > Moreover, AAEs and WAEs do not have any form of tractable density, so if they were to minimize the KL divergence between the aggregated posterior and the prior, it would have to be done adversarially. This fact in itself represents a substantial difference from CEFs, which can optimize an unbiased estimate of the objective.
> > > > > > >
> > > > > > > Lastly, AAEs/WAEs don't actually attempt to minimize the KL divergence. In reality, AAEs attempt to minimize the Jensen-Shannon divergence in $z$-space adversarially, and the more general WAE framework is an attempt to minimize the Wasserstein distance in $x$-space. Both goals suffer from the aforementioned sample complexity issues [C].
> > > > > > >
> > > > > > > ### Interpretation
> > > > > > >
> > > > > > > WAE models, including AAEs, do not share the interpretation of jointly learning a base measure (via the L2 term) and maximizing likelihood (via the likelihood/KL) on the manifold. From the WAE perspective, the L2 term minimizes the Wasserstein distance, and the divergence term is a constraint. This delineation is a critical conceptual difference from CEFs, for which the reverse is true: the L2 term is the constraint/regularizer, and the divergence term trains the likelihood.
> > > > > > >
> > > > > > >
> > > > > > > We don't insist that CEFs are better, but they are quite different, and we believe both lines of work are interesting in very distinct ways. CEF training is much more comparable to other injective flow models, some of which also use reconstruction losses to learn the manifold. Our work builds on these by structuring the flow for tractable densities; the L2 term is neither our focus nor our contribution.
> > > > > > >
> > > > > > > ## Minor Note on Continuous Pushforwards
> > > > > > >
> > > > > > > > "This is not true as well, push forward operation will not necessarily preserve the topology, it is more like a problem specific to the bijective flow (flow function is a homeomorphism) [3]."
> > > > > > >
> > > > > > > We apologize for any ambiguity on this front. We do not claim that continuous pushforwards preserve the support's topology; we claim they preserve its *connectivity*.
> > > > > > >
> > > > > > > If the domain is connected, the image under a continuous map $f$ must also be connected. Furthermore, the support of the pushforward $f_*\mu$ must be equal to the image; otherwise, $f(\mathbb{R}^m) \setminus \text{supp}(f_*\mu)$ would be non-empty, and its preimage under $f$ would have to be a non-empty open set of measure zero, contradicting that the base distribution has full support. This is why any pushforward generative model must have a connected support.
> > > > > > >
> > > > > > >
> > > > > > >
> > > > > > > [A] Makhzani, Alireza, et al. (2015). Adversarial autoencoders. arXiv preprint arXiv:1511.05644.
> > > > > > >
> > > > > > > [B] Tolstikhin, Ilya, et al. (2017). Wasserstein auto-encoders. arXiv preprint arXiv:1711.01558.
> > > > > > >
> > > > > > > [C] Arora, S., Ge, R., Liang, Y., Ma, T. & Zhang, Y.. (2017). Generalization and Equilibrium in Generative Adversarial Nets (GANs). Proceedings of the 34th International Conference on Machine Learning, in Proceedings of Machine Learning Research 70:224-232 Available from http://proceedings.mlr.press/v70/arora17a.html .

---

> > > > > > > > ### Comment · Reviewer_wnDz · 2021-08-25
> > > > > > > > **Thanks for the reply**
> > > > > > > >
> > > > > > > > Thank the author for the reply.
> > > > > > > > I agree with the point that the auto-encoder-style model cannot compute exactly the relative density value. That seems a benefit of the proposed model.  The only experiment that shows such benefit is Figure 2c. Could the author further describe how Figure 2c is visualized? Since I saw in the WGAN (which doesn't have density) comparison experiment, a similar style plot is given.
> > > > > > > >
> > > > > > > > I think if the benefit comparing to other generative models is that the proposed method can show relative density, more experiments need to be done in this track.
> > > > > > > >
> > > > > > > > Additionally, for image modeling, maybe CelebA is still a challenge, what about simpler datasets like MNIST?

---

> > > > > > > > > ### Author Response · Authors · 2021-09-01
> > > > > > > > > **Plotting Details, Density Evaluation, and MNIST**
> > > > > > > > >
> > > > > > > > > We’re glad you agree that exact computation of the density on the learned manifold is the main strength of the work. Thank you for your continued correspondence.
> > > > > > > > >
> > > > > > > > > > Could the author further describe how Figure 2c is visualized? Since I saw in the WGAN (which doesn’t have density) comparison experiment, a similar style plot is given.
> > > > > > > > >
> > > > > > > > > Thank you for raising this question - we will make this explicit in the final version. In both Figure 2c and the [WGAN experiment](https://sites.google.com/view/cefs-neurips2021/home "CEFs NeurIPS 2021 Responses"), CEF architectures were used, making the model’s density tractable and easy to visualize.
> > > > > > > > >
> > > > > > > > > The goal of the WGAN experiment was to compare our joint loss to Wasserstein training, so as noted in the experiment details in the previous response, we held the architecture constant between the two models. If a more general generator architecture were used, its density would not be tractable and we would be unable to visualize them for comparison.
> > > > > > > > >
> > > > > > > > > > I think if the benefit comparing to other generative models is that the proposed method can show relative density, more experiments need to be done in this track.
> > > > > > > > >
> > > > > > > > > We agree that showing the relative density on the manifold is important, but note a couple of difficulties in crafting more experiments:
> > > > > > > > > - There aren’t any other models to compare to; our model is the only one we know of that learns low-dimensional manifolds and has tractable densities at scale.
> > > > > > > > > - We cannot evaluate using likelihoods because different models will learn different manifolds and hence have incomparable densities. In high dimensions, we then have to use more generally applicable metrics like FID and those proposed by [1], but these do not exactly tell us the quality of the learned density.
> > > > > > > > >
> > > > > > > > > As a result, we must use low-dimensional visualizations to see the quality of the density. The Gaussian mixture on a plane from the Wasserstein comparison is one such example; we will include more analysis on this example in the paper. In [further investigations](https://sites.google.com/view/cefs-neurips2021/home "CEFs NeurIPS 2021 Responses"), we have found that a backbone with rational quadratic coupling layers [2] provides a cleaner density wherein the modes are not visibly connected.
> > > > > > > > >
> > > > > > > > > > Additionally, for image modeling, maybe CelebA is still a challenge, what about simpler datasets like MNIST?
> > > > > > > > >
> > > > > > > > > Thank you for the suggestion. We have uploaded some [MNIST samples](https://sites.google.com/view/cefs-neurips2021/home "CEFs NeurIPS 2021 Responses") from the joint CEF model. Since the response period is closing soon, we have only had time to compare to the closest baseline we can find [3], but we will include samples from our own baseline with an FID score comparison in the final paper.
> > > > > > > > >
> > > > > > > > >
> > > > > > > > > [1] Muhammad Ferjad Naeem, Seong Joon Oh, Youngjung Uh, Yunjey Choi, Jaejun Yoo (2020). Reliable Fidelity and Diversity Metrics for Generative Models. In Proceedings of Machine Learning Research 119:7176-7185. Available from https://proceedings.mlr.press/v119/naeem20a.html.
> > > > > > > > >
> > > > > > > > > [2] Conor Durkan, Artur Bekasov, Iain Murray, George Papamakarios (2019). Neural Spline Flows. In Advances in Neural Information Processing Systems 33.
> > > > > > > > >
> > > > > > > > > [3] Abhishek Kumar, Ben Poole, Kevin Murphy (2020). Regularized Autoencoders via Relaxed Injective Probability Flow. In the Proceedings of the 23rd International Conference on Artificial Intelligence and Statistics

---

### Official Review · Reviewer_YERV · 2021-07-16

**Rating:** 7
**Confidence:** 4

**Summary:**

Many datasets of interest that are modelled with normalizing flows are assumed to lie on a lower-dimensional manifold embedded in a higher-dimensional ambient space. Standard normalizing flows that are restricted to learn a *bijective* transformation are not well suited for modelling such data, as the target transformation between the lower-dimensional manifold space and the ambient space (in which said manifold is embedded) is *injective*.

In theory, an *injective flow* can be defined, where we'd parametrize an injective transformation that would only be invertible *for points on the manifold*. In practice there are two problems with this, however. First is that historically there's been much less work on designing flexible *injective* transformations, hence we don't have the same arsenal of invertible layers as we do for bijective flows. Second, the change-of-variable formula for an injective transformation includes a $\det J_f^T J_f$ term, where $J_f$ is the (rectangular) Jacobian matrix of the injective transformation $f$. This determinant, even if tractable, is typically overly expensive to compute exactly.

Authors of this work make an observation that in the bijective flow literature the game has always been to design parametric transformations $f$ that are restricted to be invertible, but *also* ones that have and easy-to-compute $\det J_f$. In line with this, authors propose a class of transformations that would have an easy-to-compute $\det J_f^T J_f$, specifically *conformal embeddings*. A conformal embedding $f$ is a transformation that preserves *angles*, but more importantly has a property that $J_f^T J_f$ is a scalar multiple of the identity, the determinant of which is simply the product of diagonal terms (i.e. an exponent of said scalar).

Authors consider a few examples of conformal embeddings (translation, scaling, an orthogonal transformation, zero-padding, etc.), which could be stacked in the same way as invertible layers are in a bijective flow. In addition, authors consider *piecewise* conformal embeddings, which are conformal *almost everywhere* (i.e. for all points except for a subset of measure zero), and can increase the expressivity of the otherwise simple conformal embeddings.

By using the conformal embeddings as injective transformations authors come up with the first injective flow with a tractable and *cheap-to-compute* density on the learned manifold. This allows said density to be used as a part of the training objective, even though an additional regularization term still needs to be included in the objective to prevent degenerate solutions. The method demonstrates excellent results on a synthetic 3D problem where the target density lies on a sphere, as well as the CelebA image dataset, where the visual fidelity of the samples is improved in comparison to the more expressive baseline flow.

**Limitations And Societal Impact:**

Authors carefully discuss both the limitations of the method and its potential societal impact.

**Main Review:**

What an interesting idea, and what a well-written paper --- very much enjoyed reading it! It could be one of these papers that kick-starts a new sub-field within the normalizing flow community, one where researchers would be working on designing novel types of expressive conformal embeddings.

Authors motivate the proposed method very well, taking the reader on a journey that arrives at the conformal embeddings as almost the one logical solution, which of course it might only be in retrospect.

The mathematical exposition is excellent: authors manage to keep things precise without drowning the reader in equations.

In my view the weakest part of the paper is the experiments. While what is done is definitely enough to get the reader interested, it's probably not sufficient to get an idea of whether the method is ready for real-world use. A comparison with alternative injective flow implementations would be of great help, as well exploring the hyper-parameters of the method (like the latent dimensionality) and additional datasets. In similar work datasets produced by sampling from a generative model with a lower-dimensional latent space (itself trained on real data) are sometimes used, where  the "real" intrinsic dimensionality of the dataset is known.

Nevertheless, I think the paper will be of great interest to the community, and the theoretical part is sufficiently novel to give authors some leeway in regards to the empirical part. I recommend acceptance, but also encourage authors to focus their efforts on the experiments for future iterations of the paper.

Minor points:
- Line 164: while it might indeed be trivial that zero-padding is conformal, I suggest including it in Table 1 for completeness (and perhaps writing a sentence summarizing the proof of the transform being conformal). This is the part that allows the flow to change dimensionality, hence it's worth being particularly precise here.
- In two places authors refer to a normalizing flow (I think?) as a "stump": this doesn't seem to be defined anywhere, and I've not seen this terminology before.
- Table 3: it might be useful to have another baseline where the "Sequential Baseline" would be run with $g$ that would have the number of parameters that is similar to $g$ used in CEF, just to make the runs more comparable.

---

I thank the authors for their response. These additional results and clarifications in the exposition will definitely strengthen the paper. Having read the other reviews/responses, I maintain my recommendation to accept the paper.

**Time Spent Reviewing:**

7

---

> ### Author Response · Authors · 2021-08-10
> **Response to Reviewer YERV**
>
> Thank you for the review! We are thrilled that you found our paper well-written and think the ideas will be of great interest for the community.
>
> You asked about (1) comparing to existing injective flow baselines, (2) exploring the dimension of the learned manifold, and (3) training on samples from another generative model with known latent dimension. Thank you for bringing these up - we address all of them in the general comment.
>
> Thank you for the minor suggestions as well.
> - We will add a short explanation showing that zero-padding is conformal. In short, the Jacobian is $J = \begin{pmatrix} I \\\\ \mathbf{0}\end{pmatrix}$, a "rectangular identity matrix", so $J^T J = I$, satisfying the conformal condition.
> - We will change the "stump" terminology to something more standard; we meant a small Glow model.
> - We trained a Small Sequential Baseline with a smaller injective component such that the parameter count was close to our CEF model. This achieved a poorer FID than our models and the other baseline: the results are visible [here](https://sites.google.com/view/cefs-neurips2021/home "CEFs NeurIPS 2021 Responses").

---

### Official Review · Reviewer_3bMb · 2021-07-19

**Rating:** 6
**Confidence:** 3

**Summary:**

This work proposes to leverage conformal embedding flows for modeling probability distributions on low-dimensional manifolds. Differing from other manifold-based flows, the proposed methods maintain tractable densities for low-dimensional latent space. The unique contribution enables fast sampling, invertibility for inference, and efficient likelihood estimation, which inherits from normalizing flows.

Existing approaches share the drawback that exact densities cannot be calculated through the dimensionality reducing step. In the paper,
the authors present an approach that offers tractable density estimation and is straightforward to train by constraining this reduction to the class of conformal embeddings.

**Ethics Review Area:**

["I don’t know"]

**Limitations And Societal Impact:**

No potential negative societal impact

**Main Review:**

Followed by recent work on manifold flows, the paper poses an attractive research direction for normalizing flows.  Overall, the paper is well-written and easy to follow. The mathematical derivations are rigorous and careful so I did not see any unclear part in the thought process. Through two experiments, the authors demonstrate the idea clearly and also give a detailed discussion about the current limitation and future direction, which would be helpful. I just have few concerns listed as follows:

1. The experiments are not very strong. I think it would be better if the authors could show more synthetic examples, for example, a more challenging one with complex manifolds. For the synthetic example, there are no other baselines for comparison such that we may not get a straightforward insight why the proposed methods are better.  For the image example, could you show more examples, maybe from a simpler one with less computational cost?  To compare the performance of generative models, I know FID is commonly used but recently some other metrics are proposed, for example, density and coverage metrics [1], which may help to evaluate the performance better.

[1] "Reliable fidelity and diversity metrics for generative models." In International Conference on Machine Learning, pp. 7176-7185. PMLR, 2020.

2. The baseline and ablation study.  The authors discussed two important baselines [3] and [27], is that possible to compare them quantitatively with some experiments, even toy examples?  It might be helpful to highlight the original contribution and unique strength of the proposed flows. As for the size of low-dimensional space, have you investigated the effect of various latent dimensionality? It is interesting to see the performance if you have the results in experiments.

3. Implementation and details. I expect to see the implementation if the authors would like to provide the code in the supplementary materials. Unfortunately, I can not reproduce the code and see more details. I just worried about the efficiency, scalability, and flow expressivity as the authors mentioned.  For the flow-based architecture, the authors only use the affline coupling layers in RealNVP and Glow, does it get improvement if some advanced flow-based models, like neural spline flows, are used?  Is the easy to incorporate the conformal embedding operator into other flow-based models? If so, that would increase the general applicability of this method.




**Time Spent Reviewing:**

9.5

---

> ### Author Response · Authors · 2021-08-10
> **Response to Reviewer 3bMb**
>
> Thank you for your review - we are pleased that you found the research direction interesting and the paper well-written.
>
> 1. You expressed a few concerns about the experiments:
>
>    a) Our synthetic example was not compared to other approaches.
>
>       We treated the synthetic example as a proof-of-concept to show that end-to-end maximum likelihood training was feasible with the restriction to conformal embedding layers. Other approaches, which are architecturally a superset of CEFs, would no doubt work, but do not admit efficient end-to-end maximum likelihood training.
>
>    b) There were not enough image examples.
>
>       We have now added more images examples on a synthetic distribution produced from a GAN manifold of known dimensionality. Please refer to the general comment.
>
>    c) FID scores were the only comparison metric for generative quality.
>
>       Your suggestion to use density and coverage in addition to FID was extremely helpful. We have now evaluated all the trained image models by first converting generated images to feature vectors with an Inception model, then evaluating density and coverage metrics on the feature vector distributions (analogous to the way FID scores are computed). Our Sequential CEF achieved the highest density (realism), whereas the baseline had the highest coverage (diversity). This supports our observation in the paper that the CEF images were visually sharper and more realistic than the baseline. These will be added to the paper.
>
>     | Model | Density | Coverage |
>     | -------- | ---------- | ------------- |
>     | Seq. Baseline |  0.1444+-0.0037 | 0.0411+-0.0016 |
>     | Seq. CEF | 0.1829+-0.0021 | 0.0347+-0.0010 |
>     | Joint CEF | 0.1303+-0.0035 | 0.0233+-0.0001 |
>
> 2. You were interested in seeing comparisons to [3] and [27], as well as the effect of changing the learned manifold's dimensionality.
>
>     As we clarify in the general comment, the Sequential Baseline we compared against is actually [3] (we will make this more clear in the revised paper). We have included a brief exploration of the CEF's performance as the learned manifold's dimension is changed, and will include any further results in the final paper.
>
> 3. You requested that we share our implementation of CEFs, and asked about incorporating architectures other than RealNVP and Glow.
>
>     We have added a preliminary, anonymized Github repo [here](https://github.com/cefs-neurips/cefs-draft "CEFs Draft Repo").
>     There is no constraint on what type of architecture can be used to learn the density on the manifold in a CEF - we used Glow since it is a simple and popular benchmark for image domains. The conformal embedding component $g$ is independent of the bijective component $h$. As you say, this gives CEFs very general applicability, since any normalizing flow architecture with tractable log-densities can be combined with a conformal manifold learner in our framework.

---

> > ### Comment · Reviewer_3bMb · 2021-09-01
> > **Thanks to the Authors for Their Response**
> >
> > I would like to thank the authors for their response to my question. I will keep my original scores

---

### Author Response · Authors · 2021-08-10
**General Response**

We are grateful to all reviewers for their feedback. We're encouraged that the work was uniformly received as well-written and original.

While there was enthusiasm about the research direction and theoretical component (**3bMb**, **YERV**), all reviewers suggested further experiments:

1. Reviewer **3bMb** suggested evaluating on more complex synthetic manifolds and more image examples. Likewise, reviewers **YERV** and **wnDz** suggested training on manifolds generated by an alternate generative model with a low-dimensional latent space.
    To address these comments, we are running experiments on a dataset derived from a class-conditional StyleGAN2-ADA [A] model trained on CIFAR-10. For a few different manifold dimensions $m$, we fixed all but $m$ latent variables and generated 10000 random images on the GAN's manifold. [Preliminary results](https://sites.google.com/view/cefs-neurips2021/home "CEFs NeurIPS 2021 Responses") using a conformal embedding flow (CEF) with corresponding latent dimension $m$ show it can learn these complex synthetic manifolds with good visual fidelity. We will add the full suite of completed experiments with the CEF and baselines to the paper.

2. Reviewers **3bMb** and **YERV** asked about investigating the latent dimensionality of the model. [Here](https://sites.google.com/view/cefs-neurips2021/home "CEFs NeurIPS 2021 Responses") are some preliminary results showing what happens when we vary the dimension by a power of 2. The dimensionality in the original manuscript is roughly the minimal one with reasonable quality on CelebA. We will add the finished results to the next copy. We will also continue to investigate the latent dimensionality on the StyleGAN2-ADA synthetic dataset and update the final version.

3. Reviewers **3bMb** and **YERV** also asked about comparing to baselines from other injective flow work. We realize upon rereading that we were not clear in the submission: the baseline shown *is* the method from [3], just scaled down to make experiments practical for us. The other comparable baseline from [27] is the same method as [3] with an architectural improvement, but at the time of our submission was unpublished with no code available, so we did not attempt to recreate their work. There are not many other examples of injective flow approaches that learn arbitrary manifolds. Reference [28] is perhaps the next closest idea, but is not left-invertible, leading to an approach more akin to an autoencoder, and again we were unable to find any reference implementation. We will clarify this situation in the paper with some added discussion.

The updated experimental results along with a draft of the codebase can found on [this page](https://sites.google.com/view/cefs-neurips2021/home "CEFs NeurIPS 2021 Responses"). All of these will be added to the final version.

[A] Karras, T., Aittala, M., Laine, S., Lehtinen, J. & Aila, T.. (2020) Training Generative Adversarial Networks with Limited Data. Advances in Neural Information Processing Systems 33.

---

### Decision · Program_Chairs · 2021-09-27

**Decision:**

Accept (Poster)

**Comment:**

The submission proposes to use conformal mapping to implement injective flow to map into a lower dimensional embedding space.
The proposed idea is original, well explained, and well motivated. While the experiments were lacking in the initial submission, the authors have properly engaged with the reviewers: they offered the appropriate additional experiments to improve the paper and address reviewers concerns.

I recommend this paper for acceptance.